# In silico modeling guides identification of novel *JAK1* variants associated with immune dysregulation

Marie Jeanpierre [1,16], Orianne Debeaupuis [1,2,16], Camille Brunaud [1], Judith Yancoski[3], Quentin Riller [1], Jerome Hadjadj[1,4], Marie-Claude Stolzenberg[1], Giselle Villarreal[5], Marie Martha Katsicas [5], Mariana Villa[6], Joao Farela Neves [7,8], Jean-Louis Stephan[9], Cédric Léonard[10], Estibaliz Lazaro[11], Jonathan Ciron [12], Charlotte Boussard[1], Fabienne Mazerolles[1], Aude Magerus[1], Pelle Olivier [1], Cecile Masson[13], Yohann Schmitt[13], Benedicte Hoareau[14], Angélique Vinit[14], Bénédicte Neven[1,15], Pierre Quartier[1,15], Herve Isambert[2], Matías Oleastro[6], Silvia Danielian[3], Marianna Parlato [1,17 ✉] & Frederic Rieux-Laucat [1,17 ✉]

## Abstract

**Characterization of primary immune dysregulations and deficiency disorders caused by hyperactivating variants of the JAK/STAT pathway highlighted its crucial role in immune cell development and response. To systematically evaluate pathogenic *JAK1* variants, we developed a structure-based predictive framework adapting AlphaFold2, modeling both the active and inactive conformations of JAK1. Dual-state modeling of 21,926 JAK1 variants enabled discrimination between pathogenic and benign variants based on their impact on regulatory conformation. Applying this approach to a large cohort of patients with suspected primary immune dysregulation and deficiency led to the identification of five novel variants located in key cis-regulatory and catalytic domains, with predicted gain of function activity. Ectopic expression of these variants in cell line resulted in varying levels of hyperactivation of JAK1 and multiple STATs at baseline. Furthermore, treatment of two patients with Tofacitinib suppressed JAK1 hyperactivation, normalized plasma cytokine levels and interferon signatures, and significantly improved clinical symptoms. These findings reveal diverse mechanisms of *JAK1* gain of function, expanding the clinical spectrum *JAK1* GOF, and underscore the importance of precise variant characterization for effective personalized therapy.**

**Keywords** JAK1; Protein Modeling; Immune Dysregulation; Personalized Medicine; JAK Inhibitors
**Subject Category** Immunology

## Introduction

The characterization of rare immunologic syndromes significantly expanded our understanding of the immune system regulation and function in humans. Advances in genetic analysis uncovered over 500 monogenic causes of primary immune dysregulation and deficiency (PIDD) (Bousfiha et al, 2022), deepening our understanding of lymphoid cell development, thymic education, apoptosis, and deletion of autoreactive cells (Wang et al, 2015). A landmark discovery in 1995 identified loss-of-function (LOF) variants in *JAK3* as the cause of Severe Combined Immunodeficiency (SCID) (Macchi et al, 1995), illustrating the crucial role of the JAK family in lymphoid development. Since then, variants in individual members of the JAK/STAT pathway have been associated to a broad spectrum of disorders, including hematological malignancies, autoinflammation and autoimmunity, all driven by dysregulated cytokine signaling (Ott et al, 2023; Hu et al, 2021).

Mechanistically, JAK and STAT proteins orchestrate signaling transduction downstream of over 50 cytokines. Upon cytokine binding to specific receptors, members of the JAK family kinases (JAK1-3 and TYK2), which are constitutively associated with these receptors, become activated through transphosphorylation of

[1]Université Paris Cité, Institut Imagine, Laboratoire d'immunogénétique des maladies autoimmunes pédiatriques, INSERM UMR1163, Paris, France. [2]Université PSL, Université Sorbonne, CNRS UMR168, Institut Curie, Paris, France. [3]Molecular Immunology, Hospital de Pediatria J P Garrahan, Buenos Aires, Argentina. [4]Sorbonne Université, service de médecine interne, Hôpital Saint-Antoine, AP-HP, Paris, France. [5]Clinical Rheumatology, Hospital de Pediatria J P Garrahan, Buenos Aires, Argentina. [6]Clinical Immunology Department, Hospital JP Garrahan, Buenos Aires, Argentina. [7]Primary Immunodeficiencies Unit, Hospital Dona Estefania, Unidade Local de Saude de São Jose, Lisbon, Portugal. [8]Comprehensive Health Research Center, CHRC, Nova Medical School, Lisbon, Portugal. [9]CHU de Saint-Etienne Hôpital Nord, Saint-Etienne, France. [10]Centre de Reference des Maladies auto-immunes Systémiques Rares de l'Est et du Sud-Ouest, Hôpital du Haut-Leveque, Pessac, France. [11]Centre de Reference des Maladies auto-immunes Systémiques Rares de l'Est et du Sud-Ouest, Hôpital du Haut-Leveque, Pessac, France. [12]CHU de Toulouse, Département de Neurologie, Pôle Neurosciences, Hôpital Purpan, Toulouse, France. [13]Genomics Core Facility, Institut Imagine-Structure Fédérative de Recherche Necker, INSERM U1163 et INSERM US24/CNRS UAR3633, Paris Cite University, Paris, France. [14]Plateforme de Cytométrie (CyPS), US-37 PASS, Sorbonne Université, Paris, France. [15]Pediatric Immunology-Hematology and Rheumatology Unit, Necker Hospital, Paris, France. [16]These authors contributed equally as first authors: Marie Jeanpierre, Orianne Debeaupuis. [17]These authors contributed equally as senior authors: Marianna Parlato, Frederic Rieux-Laucat. ✉E-mail: marianna.parlato@inserm.fr; frederic.rieux-laucat@inserm.fr

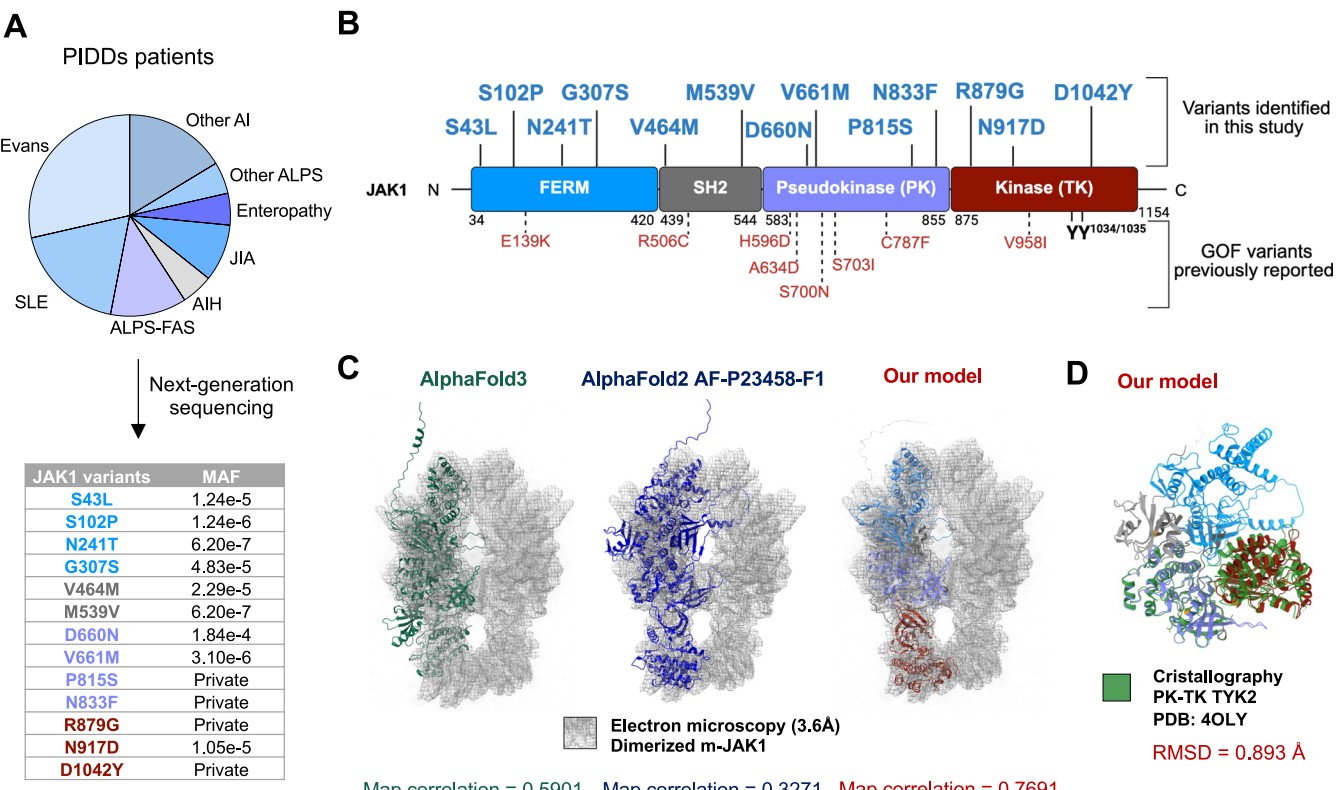

**Figure 1. In silico pathogenicity modeling prioritized JAK1 variants in patients with suspected PIDD.**

(A) Next-generation sequencing, performed on 1590 patients with immune-mediated inflammatory diseases, identified 13 rare or private *JAK1* variants. Variants are color-coded according to their location within the protein domains (Blue: FERM; gray: SH2; purple: Pseudokinase; red: Kinase). MAF Minor Allele Frequency (gnomAD v4), SLE systemic lupus erythematosus, AI autoimmunity, ALPS autoimmune lymphoproliferative syndrome, JIA juvenile idiopathic arthritis, AIH autoimmune hepatitis. (B) Schematic representation of the human JAK1 protein domains (Blue: FERM; gray: SH2; purple: Pseudokinase; red: Kinase). Previously reported GOF variants are indicated in red with dashed lines; newly identified variants are shown in blue. (C) Comparison of predicted models of the open JAK1 conformations. Left: superimposed structure of the dimerized m-JAK1 cryo-EM map with the AlphaFold3 best model (left), AlphaFold2 (middle), and the model developed in this study (right). The molecular surface from the cryo-EM map is shown as a mesh. Map correlations are displayed at the bottom. (D) Structural superposition of the closed conformation of JAK1, modeled using AlphaFold2 and colored as in (B), on the crystallographic structure of the PK-TK domains of TYK2 (PDB: 4OLI, green). The RMSD (Root-Mean-Square Deviation) between 318 pruned atoms from 4OLI and our PK-TK model is displayed at the bottom. (E) In silico pathogenicity analysis of JAK1 missense variants. A workflow combining sequence-based pathogenicity scores, functional impact scores, and computed destabilization of our open and closed conformation upon mutation (ΔΔG PyRosetta) was applied to 21,926 JAK1 missense variants. Variants were then grouped by unsupervised clustering analysis. (F) Network representation of JAK1 missense variant clustering based on in silico pathogenicity scores. Two distinct clusters emerged: Cluster 1 (7559 variants) and Cluster 2 (14,367 variants). For clarity, only 150 randomly selected variants (nodes) are displayed, each connected to its 6 nearest neighbors. Variants are colored by their origin: green (gnomAD database), red (previously reported as GOF), and blue (identified in this study's patient cohort) (see Appendix Supplementary Method 1).

conserved tyrosine residues within the activation loop of the tyrosine kinase (TK) domain. Activated JAKs then phosphorylate the intracellular tails of the receptors, creating docking sites for STAT transcription factors (STAT1-6). Once phosphorylated by the JAKS, STATs dissociate from the receptor, undergo homo- or heterodimerization with other activated-STATs, and translocate into the nucleus to drive the expression of cytokine-responsive genes often resulting in proliferation and/or differentiation. To restore basal activity, several negative feedback mechanisms finely tune this signaling circuit (Morris et al, 2018). The Suppressor Of Cytokine Signaling (SOCS) family of proteins, directly inhibit the catalytic activity of JAKs through ubiquitination, competition and degradation of JAKs. The Protein Tyrosine Phosphatases Non-receptor (PTPN) family of tyrosine phosphatase regulates the cascade by dephosphorylation of the different actors.

JAK1 is composed of four domains: the Four.1 protein Ezrin Radixin Moesin (FERM) and Src-homology 2 (SH2) domains, which are responsible for receptor binding, the pseudokinase (PK) domain, which modulates the activity of the tyrosine kinase (TK) domain, which, in turn, catalyzes the transfer of phosphate from ATP to tyrosine-containing protein (Morris et al, 2018). To date, eight gain of function (GOF) *JAK1* variants have been described in families with autosomal dominant autoinflammatory-related disease, including allergies and dermatitis, along with eosinophilic disorder and hyperimmunoglobulin E syndrome (HIES) (Takeichi et al, 2022; Gruber et al, 2020; Horesh et al, 2024; Del Bel et al, 2017; Fayand et al, 2023).

These variants were primarily located in the PK domain, which is also a known hotspot for somatic GOF variants (Vainchenker and Constantinescu, 2013), highlighting its critical regulatory role. Structural studies by Lupardus et al (Lupardus et al, 2014) on TYK2, revealed that the PK tandem domains adopt a back-to-back configuration, supporting a model where physical interactions between these two domains are required for negative regulation of JAK activation. This model was also validated by structural studies of the PK-TK modules in TYK2 and JAK2, showing that the addition of the PK fragment inhibited the constitutive activity of the TK domain (Shan et al, 2014; Lupardus et al, 2014). Until recently, our understanding of the structure and activation mechanisms of full-length JAK relied on extrapolations from structural studies of monomeric JAK fragments. Only in 2022, the structure of the mouse full-length JAK1 bound to the IFNλR1 receptor was resolved by cryo-electron microscopy, confirming that the PK domain folds onto the TK domain in a regulatory manner (Glassman et al, 2022).

Mechanistically, JAK1 exists in a dynamic equilibrium between inactive "closed", where the PK domain binds to the TK domain, and active "open" conformations that are stabilized upon receptor engagement and JAKs dimerization following cytokine stimulation (Lupardus et al, 2011). GOF variants are thought to disrupt this equilibrium. Yet, the molecular mechanisms underlying many of these variants remained unknown and may also involve complex crosstalk with other JAK family members. Therefore, detailed insights into JAK1 structural dynamics are key to understand the impact of these variants at the molecular level, how they affect cis and/or transregulation among JAKs, as well as to refine drug targeting strategies.

In this study, we combined in silico modeling of JAK1 structure with predictive protein modeling tools to assess the impact of *JAK1* variants. This approach, further validated by functional assay, both in vitro and ex vivo, allowed the identification of novel *JAK1* GOF variants in five patients presenting with broad clinical manifestations, including autoimmune hepatitis and recurrent infections, and defined targeted therapeutic strategies.

## Results

### Dual-state structural modeling of 21,926 *JAK1* variants identifies potential pathogenic candidates

Sporadic reports of *JAK1* GOF variants with PIDD associated to autoimmunity or inflammation have significantly advanced our understanding of the molecular mechanisms underlying JAK/STAT activation, which is key for guiding targeted therapies (Ott et al, 2023). Yet, assigning pathogenicity to newly identified variants remains challenging, as functional validation is time-consuming and not feasible in routine clinical settings. Thus, we applied a structure-guided approach that leverages JAK1's dynamic conformational landscape to differentiate neutral variants from those potentially driving GOF phenotypes. In our cohort of 4000 patients with suspected PIDD, next-generation sequencing (NGS) was performed on 1590 individuals presenting with autoimmune symptoms before 18 years of age (Fig. 1A), revealing 13 rare or previously unreported *JAK1* variants (Fig. 1A), distributed across various domains of JAK1 (Fig. 1B). To assess their potential impact on protein structure, we adapted the AlphaFold2 algorithm (Appendix supplementary Method 1) to model JAK1 in both open (active) and closed (inactive) conformations (Fig. 1C,D; Appendix

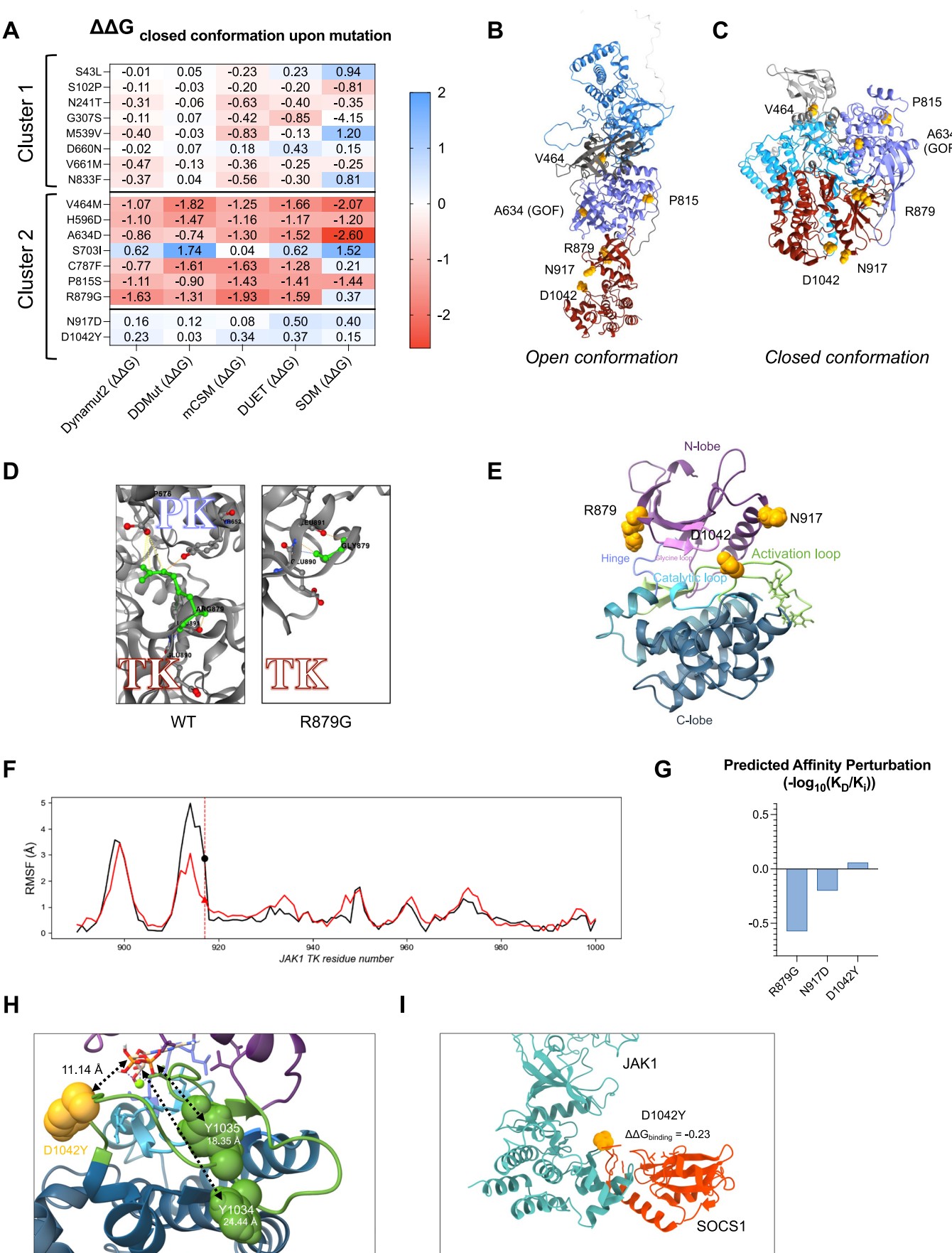

◀ **Figure 2. In silico modeling revealed diverse mechanisms of kinase dysregulation caused by predicted damaging *JAK1* variants.**

(A) Stability perturbation (ΔΔG) driven by variants was calculated on closed conformation using five different methods (see Appendix Supplementary Methods). SDM computes raw site-directed mutagenesis effect; mCSM predicts protein destabilization and protein–protein binding affinity changes. DUET combines SDM and mCSM outputs, while DDMut and Dynamut2 assess vibrational entropy changes alongside alterations in protein stability and flexibility. (B, C) Structural models of JAK1 open (B) and closed (C) conformations with mutated amino acids positions shown in yellow. Domains are color-coded as in Fig. 1B. (D) Comparison of interaction loss upon p.R879G variants with the wild-type (WT) shown in the left panel and p.R879G in the right panel. (E) Predicted active site of JAK1 in the closed conformation, with functional sites color-coded and annotated. Variant localizations are indicated with yellow spheres. The activation loop (green) folds over the catalytic loop (light blue), leading to its inhibition. ATP and ADP docking to the site pocket proved unfeasible. (F) Root-mean-square fluctuation (RMSF, in Angstroms) line plots from molecular dynamics simulations comparing the p.N917D variant (red) and the wild-type (black) in separated assays. RMSF measures the deviation from the original coordinate of the residue and is performed for each structure independently. The residue position is indicated by a red dashed line; wild-type residue by a black dot and mutated residue by a red triangle. (G) Histogram illustrating predicted ligand-induced protein destabilization score ($-\log_{10}(K_D/K_i)$) from mCSM-lig analysis for ADP binding per variant. (H) Zoom on the activation loop in the open JAK1 conformation bound to ADP. WT tyrosines ($Y^{1034/1035}$) working as phosphorylation acceptors are shown as green spheres, while the D1042Y mutant is shown as a yellow sphere. Dashed lines indicate (in Angstroms) the distances between each tyrosine to the center of the ADP binding pocket, highlighting the unexpected spatial symmetry between the mutant and the native phosphorylation sites. (I) Predicted binding interaction between SOCS1 protein and the TK domain of the $JAK1^{D1042Y}$ mutant modeled using AlphaFold2. The destabilizing effect of the D1042Y substitution on this interaction is quantified by ΔΔG. Source data are available online for this figure.

Fig. S1A) as conformational state is a key regulator of JAK1 activity (Glassman et al, 2022).

The open-state JAK1 model mapped closely with the recently resolved cryo-EM structure of the full-length mouse JAK1 complexed with IFN-λ receptor (Glassman et al, 2022), outperforming AlphaFold2 & 3 predictions (Fig. 1C; Movies EV1–3; Appendix Fig. S1B). While no closed-state conformation of JAK1 has been reported to date, our model of the inactive conformation aligned well with the crystallographic structure of TYK2 PK-TK folding (Fig. 1D; Appendix Fig. S1B) (Lupardus et al, 2014).

We developed an in silico pipeline, integrating multiple structural and sequence-based metrics. Using PyRosetta, we estimated the conformational destabilization (ΔΔG) caused by each variant in both open and closed states. This parameter was combined with sequence-based pathogenicity score (CADD, AlphaMissense), evolutionary score (ESMb-1), and functional prediction scores (Polyphen2, SIFT, MutFunc). We applied this pipeline to 21,926 possible JAK1 variants, generated by substituting each of the 1154 amino acidic residues with all 19 alternative amino acids (Fig. 1E). Based on their predicted pathogenicity profiles (Fig. EV1A), these variants segregated in two clusters, Cluster 1 (7559 variants) and Cluster 2 (14,367 variants) (Fig. 1F; Dataset EV1). Interestingly, Cluster 1 was enriched for *JAK1* variants frequently reported in gnomAD (214/289) (green) (Fig. EV1B), suggesting that this cluster may represent variants with low pathogenicity threat. In contrast, Cluster 2 exhibited a lower mean ESMb1 evolutionary score (−10.75 vs. −4.47) and a higher mean CADD score (26.68 vs. 20.99) compared to Cluster 1. This trend was consistent across other scores, including MutFunc, AlphaMissense, Polyphen2, and ClinPred. In addition, Cluster 2 showed a substantially lower mean ΔΔG for the closed conformation (−6.36 vs. −2.05), suggesting a greater destabilizing impact (Fig. EV1A). Overall, the integration of these scores points to a higher likelihood of pathogenicity, which may result either in a gain or loss of function.

To validate our clustering, we overexpressed the 13 JAK1 variants identified in our cohort of patients in JAK1-deficient U4C cells, along with a luciferase reporter driven by the ISRE promoter to assess STAT1 and STAT2 transcriptional activity as a proxy of JAK1 activity. The previously reported GOF variant, p.A634D, was included as a positive control. Notably, only the five variants (blue) that clustered with known GOF variants (red) in Cluster 2

exhibited increased STAT transcriptional activity at baseline and upon IFN-α stimulation (Fig. EV1C). These findings support their classification as potential GOF for further investigation.

## In silico JAK1 modeling reveals diverse mechanisms of kinase dysregulation

To further assess the impact on protein conformation or binding of the five candidate variants in Cluster 2, we first computed their thermodynamic stability shifts (ΔΔG) using five distinct tools: site-directed mutagenesis (Topham et al, 1997; Worth et al, 2011) (SDM, structural analysis), Cutoff Scanning Matrix (Pires et al, 2014a) (mCSM, binding analysis), DUET (Pires et al, 2014b), DDMut (Zhou et al, 2023) and Dynamut2 (Rodrigues et al, 2021) (Deep Learning) (Fig. 2A; Appendix Fig. S2). As control, we included neutral variants from Cluster 1 (Figs. 1F and EV1C) and previously described GOF variants in the PK domain, which also fall into Cluster 2 (Reported GOF: H596D, A634D, S703I and C787F, Fig. 1F). The five newly identified candidate variants were mapped onto both the open (Fig. 2B) and closed (Fig. 2C) JAK1 conformations. Notably, variants p.H596D and p.A634D mapped at the predicted PK-TK interface (Rodriguez Moncivais et al, 2023), suggesting that disruption of this interaction destabilizes the closed, autoinhibited, JAK1 conformation, leading to a loss of cis-regulation and resulting in gain-of-function. Three out of four GOF variants previously reported (Takeichi et al, 2022; Del Bel et al, 2017; Fayand et al, 2023) (p.H596D, p.A634D and p.C787F) were predicted to destabilize both the closed and open conformations of JAK1, while none of the neutral variants showed similar destabilizing effects (Figs. 2A and EV2A). In addition, p.V464M, p.P815S, and p.R879G variants were predicted to destabilize the closed conformation (Fig. 2A), as well as the open conformation (Fig. EV2A). Interestingly, for the p.V464M variant, the methionine residue shares physicochemical properties with valine and alanine, both are known to be pathogenic at this position (Appendix Fig. S3A). This suggests that allosteric perturbations may drive the instability of the p.V464M variant. The p.R879G variant exhibited a positive ΔΔG score (Fig. EV2B), indicating a shift toward the open, catalytically active conformation. This finding is consistent with the structural model, where replacing the large Arginine with the smaller Glycine (Gly879) disrupts hydrogen bonding between the kinase domain and residues P576 and Y652 in the PK domain

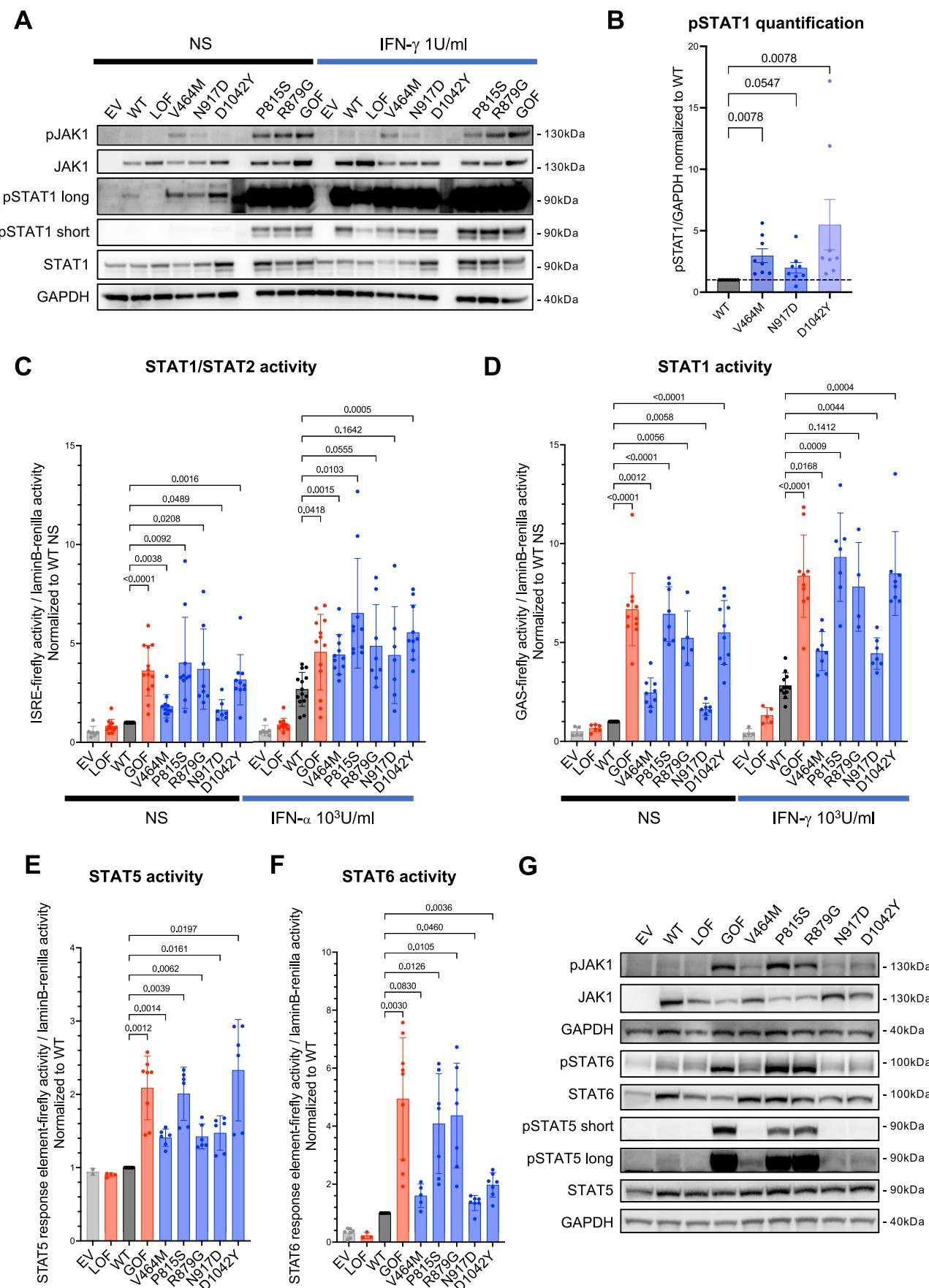

**Figure 3.  Predicted pathogenic *JAK1* variants resulted in increased JAK1 autophosphorylation and activate multiple downstream STAT pathways.**

(A) Representative Western Blot analysis of phospho-JAK1 and phospho-STAT1 in U4C cells transduced with WT or variant JAK1 constructs at baseline or after stimulation with IFN-γ (1U/ml for 15 min). (B) WB quantification of phospho-STAT1 in U4C cells transduced with WT and different JAK1 variants (p.V464M, p.N917D, and p.D1042Y) at baseline. pSTAT1 band quantification is normalized to total GAPDH and WT NS condition. $n = 8$ independent experiments. Statistical significance was determined by Wilcoxon $t$ test. $P$ values for the statistical comparisons are shown in the figure. Error bars represent the standard error of the mean. (C) ISGF3 complex (STAT1, STAT2 and IRF9) luciferase activity in U4C cells co-transfected with an Interferon Stimulated Response Element (ISRE) reporter and WT or variant JAK1 constructs at baseline or after stimulation with IFN-α ($10^3$ U/ml for 6 h); $n > 7$. Statistical significance was determined by one-way ANOVA with Tukey's multiple comparisons test. $P$ values for the statistical comparisons are shown in the figure. Error bars represent the standard error of the mean. (D) STAT1 luciferase activity in U4C cells co-transfected with a Gamma interferon activation site (GAS) reporter and WT or variant JAK1 constructs at baseline or after stimulation with IFN-γ ($10^3$ U/ml for 6 h); $n > 4$. Statistical significance was determined by one-way ANOVA with Tukey's multiple comparisons test. $P$ values for the statistical comparisons are shown in the figure. Error bars represent the standard error of the mean. (E, F) STAT5 (E) or STAT6 (F) luciferase activity in U4C cells co-transfected with a 5× STAT5 binding site (E) or a 4× STAT6 binding site (F) reporter and WT or variant JAK1 constructs; $n > 6$. Statistical significance was determined by one-way ANOVA with Tukey's multiple comparisons test. $P$ values for the statistical comparisons are shown in the figure. Error bars represent the standard error of the mean. (G) Western blot analysis of JAK1, STAT5, and STAT6 phosphorylation at baseline in U4C cells transduced with WT or variant JAK1 constructs. Representative of six experiments. Source data are available online for this figure.

(Figs. 2D and EV2C,D). The p.P815S variant showed the greatest destabilizing effect on the open conformation. This is reflected by the highest root-mean-square deviation (RMSD) (Fig. EV2E) and significant structural fluctuation, suggesting major structural alterations compared to other variants.

Collectively, these findings support the hypothesis that p.V464M, p.P815S, and p.R879G destabilize JAK1 conformations, leading to a more active structure and gain-of-function activity. While p.N917D and p.D1042Y variants showed no predicted structural change, their locations in the TK domain of JAK1 suggest they may still affect the kinase's catalytic activity (Fig. 2E). Specifically, the p.N917D variant altered the intrinsic physical characteristics of the enzyme's active site pocket. Molecular dynamic simulations of the p.N917D JAK1 variant revealed a significant reduction in root-mean-square fluctuation (measures the deviation from the original coordinate of the residue) over time, suggesting that this variant could contribute to a more rigid loop and a potentially more accessible pocket site (Fig. 2F), which is crucial for substrate and ATP binding. To assess the impact on catalytic activity of the p.R879G, p.N917D, and p.D1042Y variants, located in the kinase domain near the ATP binding site (Fig. 2E), we quantified their ADP-binding affinity. In contrast to p.N917D and p.D1042Y, the p.R879G variant significantly altered ADP affinity as indicated by its $K_D$ constant ($-\log10 K_D/K_i$) (Fig. 2G). Given that ADP release is often the rate-limiting step for many kinases, the p.R879G variant may lead to increased catalytic efficiency.

In addition, the location of the p.D1042Y variant within the JAK1 activation loop (Fig. 2E) raised intriguing possibilities. JAK1 activation was known to require robust phosphorylation at position $Y^{1034}/Y^{1035}$ within the activation loop. Various tyrosine positions were found in the activation loop in other species (Appendix Fig. S3B). Thus, the D1042Y variant could serve as a novel phosphorylation acceptor site, a hypothesis supported by the close distance between $Y^{1034/1035/1042}$ to the center of the active pocket (Fig. 2H).

Furthermore, this variant localizes at the site involved in interaction with the KIR domain of SOCS1. Based on the reported interaction between JAK1's TK domain and SOCS1, we calculated the thermodynamic impact of this variant on the stability of this interaction, which suggested it could destabilize the bond (Fig. 2I).

Overall, prediction models suggested that the identified variants could exert GOF by different mechanisms.

## Novel *JAK1* variants increased JAK1 and multiple STATs activity

To test the functional impact of the identified *JAK1* variants, we used U4C cells, a well-established model for studying JAK1 signaling (Horesh et al, 2024; Gruber et al, 2020). Since transient transfection of JAK1 is known to induce activation-loop phosphorylation, resulting in phosphorylation of JAK1 (Gordon et al, 2010), we stably transduced U4C cells with each variant. Known GOF (p.A634D) (Del Bel et al, 2017) and LOF (p.K908A) (Li et al, 2013) variants served as positive and negative controls, respectively. At baseline, we observed a trend toward increase in JAK1 phosphorylation for the p.V464M and p.N917D variants, while the p.P815S, p.R879G, and A634D GOF variants showed a pronounced increase (Fig. 3A). Assessment of STAT1, a direct JAK1 substrate, revealed elevated baseline phosphorylation across all variants, with the strongest effects seen for p.P815S, p.R879G, and A634D (Fig. 3A). Repeated quantification confirmed that even the milder variants, p.V464M, p.N917D, and p.D1042Y, exhibited a consistent, although modest, increase in STAT1 phosphorylation at baseline (Fig. 3B), supporting their classification as GOF variants with lower activity. Next, we assessed the STATs transcriptional activity using a luciferase reporter under the control of various promoters, including ISRE (STAT1/STAT2 (Fig. 3C), GAS (STAT1) (Fig. 3D), STAT5 (Fig. 3E), or STAT6 (Fig. 3F) response elements. At the basal state, overexpression of each mutant resulted in increased luciferase activity for all the tested reporters (Fig. 3C–F) correlated with increased phosphorylation of STAT1 (Fig. 3A,B), STAT5 and STAT6 (Fig. 3G). For the latter, we were able to observe phosphorylation only for variants with higher GOF activity (p.P815S and p.R879G). Of note, increased GAS and ISRE luciferase activity was also observed upon IFN-γ stimulation or IFN-α stimulation, respectively (Fig. 3C,D). While U4C cells expressing the p.P815S and p.R879G variants exhibited similar activity compared to the A634D GOF *JAK1* variant, the p.V464M, p.N917D, and p.D1042Y variants exhibited a milder GOF effect (Fig. 3).

## Variable clinical expression of immune dysregulation in patients with GOF *JAK1* variants

Patients carrying the newly identified GOF *JAK1* variants exhibited heterogeneous clinical phenotypes (Table 1 and Supplementary Case

**Table 1.** Broad spectrum of clinical manifestations in patients with predicted pathogenic *JAK1* variants compared to patients with JAK/STAT pathway hyperactivation.

| | JAK1-V464M 2002 | JAK1-P815S 2008 | JAK1-R879G 2006 | JAK1-N917D 2009 | JAK1-D1042Y 2001 | JAK1 AD GOF | STAT1 AD GOF[‡] | STAT5b AD GOF[¶] | STAT6 AD GOF[®] | SOCS1 AD LOF[‡] | PTPN2 AD LOF[$] |
|---|---|---|---|---|---|---|---|---|---|---|---|
| Year of birth | 2002 | 2008 | 2006 | 2009 | 2001 | | | | | | |
| Sex | Female | Male | Female | Male | Male | | | | | | |
| Onset | Early childhood | Neonatal | Neonatal | Neonatal | Neonatal | Neonatal-Childhood | Neonatal | Neonatal | Early childhood | Early childhood | Early childhood - young adult |
| Variant's carriers | 2 | 1 | 2 | 2 | 1 | 30 | 274 | 2 | 21 | 67 | 14 |
| Penetrance | Incomplete | Complete | Incomplete | Incomplete | ? | Incomplete | Incomplete | Complete | Complete | Incomplete | Incomplete |
| Failure to thrive/short stature | Short stature (1.40 m) | Failure to thrive, short stature (1.40 m) | Late-onset cord separation | No | Moderate intrauterine growth restriction | 43% | - | - | 43% | - | 7% |
| **Autoimmunity/inflammation** | | | | | | | | | | | |
| Enteropathy/IBD | No | No | No | No | Vomiting, false routes, gastrostomy, diarrhea, abdominal pain | 36% Diarrhea/constipation, IBD | 2% | Diarrhea | 71% | 15% IBD, 13% chronic diarrhea | 14% |
| Cutaneous diseases | Rosacea | Severe eczema and AD, necrotizing vasculitis, sclerosis | AD characterized by prurigo nodularis | Mild atopic manifestations, episodes of prurigo, skin infections | Melanoderma | 56% AD | 10% (vitiligo, alopecia, psoriasis) | 100% AD, Urticaria | 90% AD, 38% recurrent skin infections | 29% AD, 21% psoriasis, 3% alopecia | - |
| Eosinophil count | NA | Eosinophilia | NA | Eosinophilia | NA | High 43% | - | 100% High | 100% High | 19% High | - |
| Hyper-IgE | NA | High | NA | High | No | + | - | 100% High | 100% High | + | - |
| Other AI diseases | Type-1 AIH | No | No | No | Addison's disease, alopecia, vitiligo | 3% AIH, 6% Addison's disease, 3% hypothyroidism, 3% ITP | 2% AIH, 2% IPEX-like, 22% hypothyroidism, 4% T1D, 4% AI cytopenia | - | - | 15% SLE, 12% thyroiditis, 39% AI cytopenia | 7% SLE, 28% Evans |
| Other inflammatory conditions | Chalazion, rosacea | No | Episode of hyperinflammation with hepatitis, proteinuria, serositis | No | Hepatic disease (without classical autoantibodies) | 16% asthma | - | - | 62% asthma | 19% Liver disease, 10% Interstitial lung disease, 27% asthma | 14% Interstitial lung disease |
| **Immune deficiency** | | | | | | | | | | | |
| Hypogammaglobulinemia | IgA deficiency, high IgM and IgG | NA | NA | No | No | 13% Mild hypoglobulinemia | HyperIgG (20% of tested patients) | - | +/− | 12% | CVID 7% |

**Table 1.** (continued)

| | JAK1-V464M 2002 | JAK1-P815S 2008 | JAK1-R879G 2006 | JAK1-N917D 2009 | JAK1-D1042Y 2001 | JAK1 AD GOF | STAT1 AD GOF‡ | STAT5b AD GOF¶ | STAT6 AD GOF@ | SOCS1 AD LOF‡ | PTPN2 AD LOF§ |
|---|---|---|---|---|---|---|---|---|---|---|---|
| **Year of birth** | | | | | | | | | | | |
| Infection susceptibility | Cellulitis on thigh | No | Psoas-iliac abscess | Pneumonia (2 y) Cutaneous infection (3 y) Clinical bacteremia (3 y) Dental phlegmon (5 Y) Myositis paravertebal abscess *S. aureus* (9 y) Pharyngitis Adenitis CMV viremia | No | 30% recurrent infections | 98% CMC, 74% bacterial infections, 38% viral infections | - | 38% recurrent skin and 29% respiratory tract infections | 19% bacterial infections, 15% viral infections, 4% fungal infections | 14% |
| Warts | No | No | No | No | No | 13% | - | - | - | - | - |
| **Onco-hematologic features** | | | | | | | | | | | |
| Lymphadenopathy | Lymphopenia | No | Yes | Yes | No | $+/-$ | - | - | - | 12% | 14% |
| Hepatosplenomegaly | Splenomegaly | No | No | Yes | No | 10% | - | - | - | 31% | 7% |
| Multiple CFT | No | No | No | No | No | 23% | - | - | - | - | - |
| Other feature(s) | Vascular abnormalities (Raynaud's phenomenon, necrotizing vasculitis) | | | Hyperlaxity | Developmental and language impairments. Joint hyperlaxity. Genetic cerebellar ataxia (homozygous mutation SETX) | 1 patient with moderate motor impairment and learning disability, autism. | 6% aneurysms | - | Thickening of facial features, osteoporosis, pathologic fractures | Cognitive impairment, 14% psychiatric disorders | Cognitive impairment, 14% psychiatric disorders |

*NA* not assessed, *AIH* autoimmune hepatitis, *AD* atopic dermatitis, *CMV* cytomegalovirus, *IBD* Inflammatory bowel disease, *ITP* immune thrombocytopenic purpura, *IPEX* immune dysregulation, polyendocrinopathy, enteropathy, X-linked, *T1D* type 1 diabetes, *CMC* chronic mucocutaneous candidiasis, *SLE* systemic lupus erythematosus, *CVID* common variable immunodeficiency, *CFT* cystic fibrosis, *LGL* large granular lymphocyte.
§Okada et al, 2020.
¶Kasap et al, 2022.
@Sharma et al, 2024.
‡Hadjadj et al, 2025.
§Parlato et al, 2020; Jeanpierre et al, 2024; Awwad et al, 2023; Roppelt et al, 2024; Thaventhiran et al, 2020.

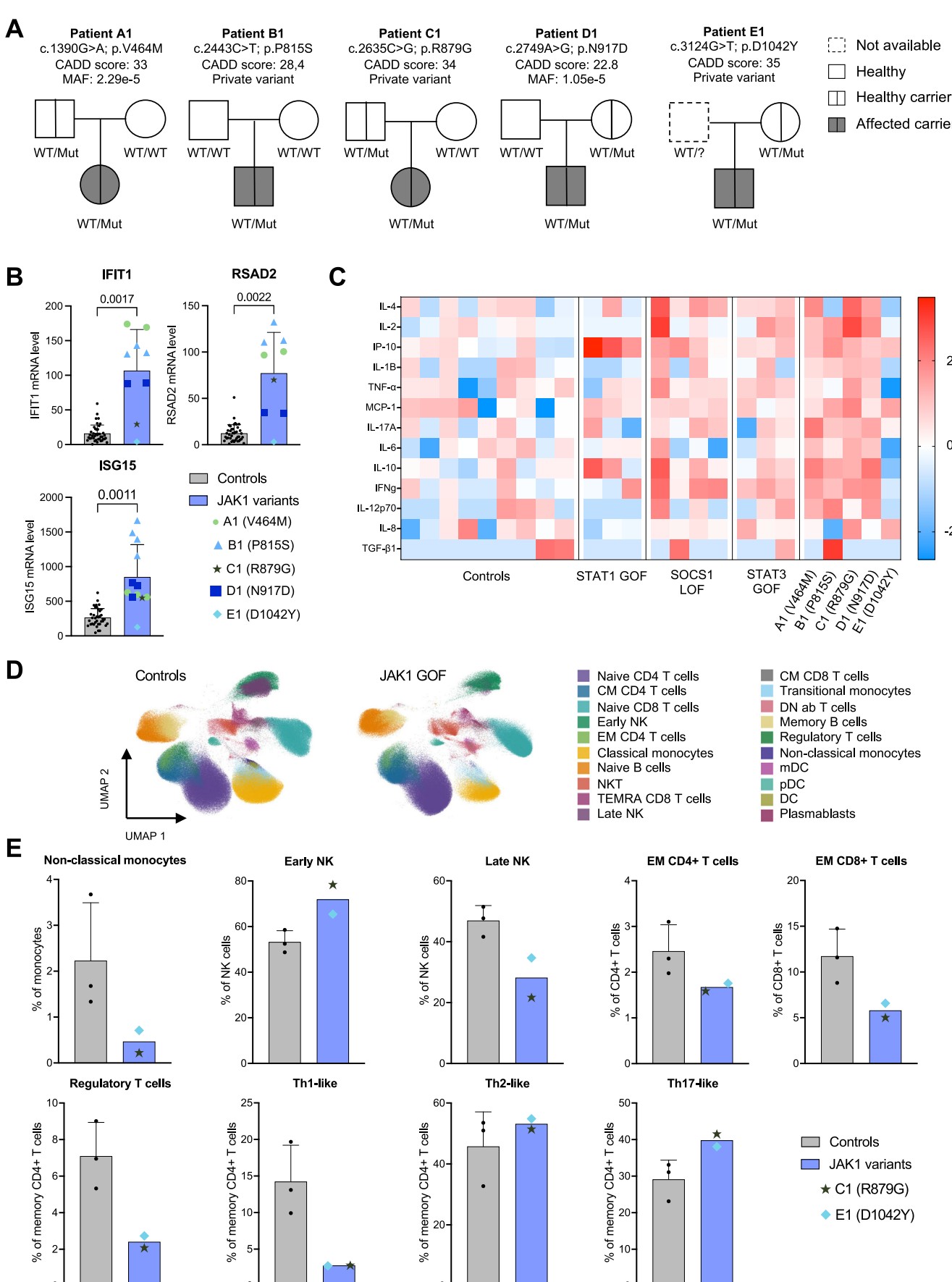

◄

**Figure 4. Increased JAK/STAT pathway activation and imbalance of immune cells development in patient's primary cells.**

(A) Pedigree of the five affected patients, indicating the frequency of the variant (MAF, gnomAD v4) and the corresponding CADD scores. Filled symbols indicate affected individuals. (B) Expression of interferon-stimulated genes (ISGs) normalized to GAPDH in PBMCs from the five patients (A1: $n = 2$; B1: $n = 3$; C1: $n = 1$; D1: $n = 2$; E1: $n = 1$) and fourteen controls. Statistical significance was determined by $t$ test (Welch). $P$ values for the statistical comparisons are shown in the figure. Error bars represent the standard error of the mean. (C) Heatmap showing cytokine concentrations in plasma samples from controls ($n = 9$), STAT1 GOF patients ($n = 3$), STAT3 GOF patients ($n = 3$), SOCS1 LOF patients ($n = 4$) and JAK1 variant ($n = 5$) carriers. The log10 for each cytokine levels were normalized to controls. (D) UMAP plots generated from thawed PBMCs from age-matched controls ($n = 3$) and patients' cells (C1, E1) by mass cytometry (CyTOF). To avoid possible visualization bias in cell proportions representation, control samples were randomly downsampled to match the total cell count of patient samples. (E) Quantification of immune cell subsets from controls ($n = 3$) and patients (C1, $n = 1$ and E1, $n = 1$), based on CyTOF analysis. Error bars represent the standard error of the mean. Source data are available online for this figure.

report), expanding the previously reported spectrum of JAK1 GOF-associated disease. Yet, these patients shared core clinical features consistent with JAK/STAT pathway hyperactivation, similar to those observed in patients with GOF variants in STAT1 (susceptibility to infections), STAT5 (atopic dermatitis), or STAT6 (hyper-eosinophilia and allergy), or haploinsufficiency of SOCS1 or PTPN2 (multi-organ autoimmunity). Patient A1, first child of otherwise healthy parents, presented selective IgA deficiency and persistent lymphopenia. The patient presented with cellulitis of the thigh of unknown origin. At the age of 10 years, the patient developed early-onset autoimmune hepatitis, leading to liver transplant. After a year of unspecific immunosuppressive therapy course, no relapse was observed. Patient B1 presented at 2 months old with severe, refractory eczema and dermatitis, alongside failure to thrive. The presentation progressively involved polyarthralgia, necrotizing vasculitis of the extremities, and cutaneous sclerosis with Raynaud's phenomenon. Patient C1 suffered from overt lymphoproliferation at the age of 13 years, along with a psoas abscess of unknown origin, severe systemic inflammation, including prolonged fever, and polyserositis. Continuous intravenous antibiotics and corticosteroids were required for management. Patient D1's clinical course was mainly characterized by recurrent infections and atopic dermatitis. A pancytopenia was reported at 6 months old (of presumed infectious origin), and infections included pneumonia at 2 years old, adenophlegmon at 5 years old, and a *Staphylococcus aureus* paravertebral infection at 9 years old. Laboratory findings revealed marked eosinophilia and elevated IgE levels. Patient E1 presented with early-onset polyautoimmunity, including vitiligo at 2 years old, teenagerhood-onset alopecia, and was diagnosed with Addison's disease at 13 years old, with positive anti-21-hydroxylase autoantibodies. Additional symptoms included enamel dysplasia and intellectual disability. By age 12, the patient developed ataxia with oculomotor apraxia type 2, attributed to a SETX (Schöls et al, 2008) gene variant. Disease progression resulted in severe motor impairment, de facto leading to wheelchair dependence and complete loss of autonomy. The complete clinical history of patients is summarized in the Supplementary Case report.

The JAK1 variants were inherited from a healthy parent for all the patients except in patient B1 where it arose de novo (Fig. 4A). Of note, all tested parents were healthy, suggesting incomplete clinical penetrance.

## JAK1 variants led to hyperactivation of the JAK/STAT pathway in patients' cells and imbalance of immune cells development

To assess the functional impact of the variants in primary cells from the patients, we first tested the expression of Interferon-Stimulated Genes (ISGs) in peripheral blood mononuclear cells (PBMCs). Elevated levels of ISG15, RSAD2, and IFIT1 transcripts

were observed in patients carrying the p.V464M, p.P815S, p.R879G, and p.N917D variants as compared to controls (Fig. 4B). Notably, the highest expression of these ISG transcripts was detected in cells from the patient carrying the p.P815S variant. These results were consistent with the functional data obtained in the ectopic expression system, where STAT1 transcriptional activity was elevated for all variants but highest for the p.P815S variant (Fig. 3).

In addition, all patients but E1 (who was receiving corticosteroid treatment at the time of sampling) exhibited elevated level of pro-inflammatory mediators in plasma (Fig. 4C). A similar pro-inflammatory profile was also observed in patients with GOF variants in STAT1 and STAT3, and LOF variants in SOCS1 (Fig. 4C). Of note, the most significantly elevated inflammatory cytokines in all patients' plasma, IP-10 and MCP-1, were also found increased in the supernatant of U4C cells transfected with the different JAK1 variants (Fig. EV3), highlighting the role of JAK1 in the production of these pro-inflammatory cytokines.

Finally, JAK1 was known to participate in signal transduction downstream of numerous cytokines, including those responsible for immune cell development and responses. To assess whether JAK1 GOF variants resulted in abnormal myeloid or lymphoid development, we performed mass cytometry-based immunophenotyping (CyTOF) in PBMCs from two patients (C1 and E1) (Fig. 4D). Compared to age-matched controls, patients carrying JAK1 GOF variants showed a decreased frequency of non-classical monocytes as well as late NK subpopulation (Fig. 4E). While the frequency of CD4$^+$ and CD8$^+$ T cells was normal (Appendix Fig. S4), there was a decrease in effector memory (EM) proportions (Fig. 4E). Within the CD4+ T-helper population, we observed a decrease in the Th1-like and Treg subsets, alongside a tendency toward elevated Th17-like cells frequency (Fig. 4E). These finding were consistent with the autoimmune manifestations observed in the patients.

## Tofacitinib treatment effectively rescued hyper JAK/STAT activity in cells expressing JAK1 GOF variants

The identification and characterization of hyperactivating JAK1 variants prompted the implementation of targeted therapy using JAK inhibitors in two patients (C1 and D1). To evaluate and guide potential treatment selection, we tested the efficacy of both mono-specific and pan-JAK inhibitors in vitro using U4C cells expressing the p.P815S and p.R879G variants. Treatment with the pan-JAK inhibitor Tofacitinib (which targets JAK3, JAK2, JAK1, and weakly TYK2) resulted in a complete loss of STAT1 phosphorylation. In contrast, the JAK1-specific inhibitor Abrocitinib and the JAK2 inhibitor CEP achieved only partial reductions (Figs. 5A and EV4). Consistently, whole blood samples from patient C1 showed a marked decrease in STAT1 phosphorylation at baseline and following IFN-α and IL-2 stimulation after therapy, in contrast to

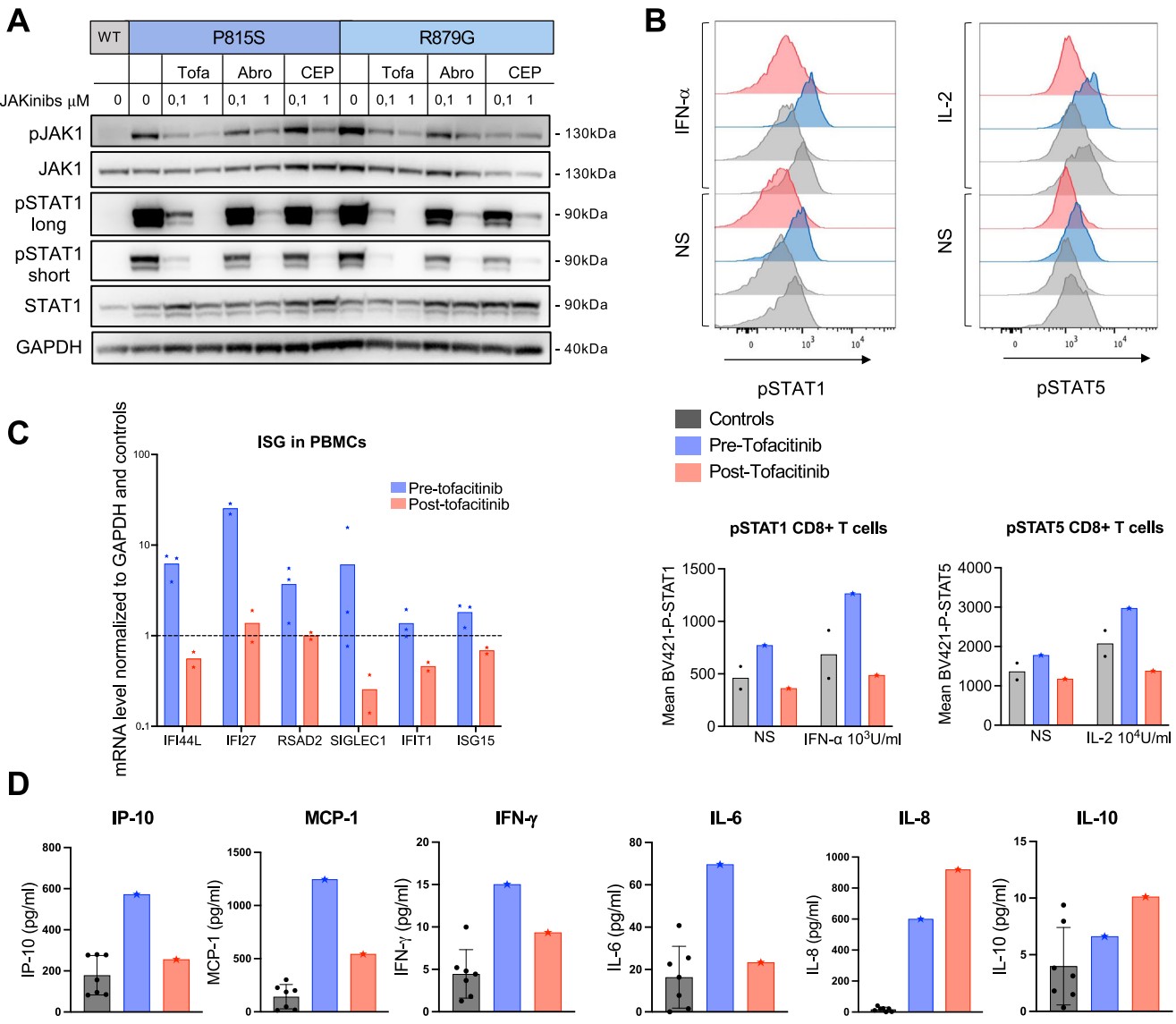

**Figure 5. Tofacitinib treatment normalizes JAK/STAT pathway activation in cells expressing *JAK1* GOF variants.**

(A) WB analysis of phospho-JAK1 and phospho-STAT1 in U4C cells transduced with WT, p.P815S or p.R879G mutants treated with Tofacitinib (pan-JAK inhibitor), Abrocitinib (JAK1 selective inhibitor) or CEP-33779 (JAK2 selective inhibitor) at 0.1 or 1 μM during 4 h. Data are representative of two experiments. (B) Representative ridgeplot showing STAT1 phosphorylation in CD8 + T cells at baseline and after 15 min IFN-α 10³ U/ml stimulation (upper panel) and STAT5 phosphorylation (bottom panel) in CD8 + T cells at baseline and after IL-2 10⁴ U/ml stimulation in whole blood from patient C1 before (blue) and after treatment with Tofacitinib (red) compared to controls (gray). Quantification of phospho-STATs is shown below. (C) ISGs expression in PBMCs from patient C1 (R879G) before ($n = 1$, blue) or after ($n = 1$, red) treatment with Tofacitinib. mRNA expression for each gene was normalized to controls ($n = 3$ before; $n = 2$ after treatment). The black line indicates gene expression in controls. (D) Plasma cytokine levels in patient C1 before (blue) and after Tofacitinib treatment (red) compared to controls (gray, $n = 7$). Error bars represent the standard error of the mean. Source data are available online for this figure.

the elevated levels seen before treatment (Fig. 5B). In addition, the transcriptional profile of interferon-stimulated genes (ISGs) in patient C1' PBMCs showed a substantial decrease after Tofacitinib treatment, with expression levels returning to those seen in control PBMCs (Fig. 5C). Similarly, Tofacitinib treatment normalized the levels of several inflammatory cytokines including IP-10 and MCP-1 (Fig. 5D), highlighting the overall efficacy of this targeted therapy.

In terms of clinical outcomes, patient B1 achieved sustained remission after Tofacitinib treatment, following a partial response to cyclophosphamide and corticosteroids. Prurigo nodularis and

perniosis were also effectively treated with Tofacitinib in patient C1 (Appendix Supplementary Case report).

## Discussion

We developed an in silico tool combining pathogenicity prediction scores and variant-induced conformational destabilization of JAK1 to assess the pathogenicity of 13 missense variants identified in a cohort of 1590 patients with immune-mediated autoimmune or

inflammatory diseases. This approach led to the identification of five novel *JAK1* GOF variants, which were further validated by functional assay showing increased JAK1 or STAT1 phosphorylation and enhanced downstream STATs transcriptional activity. The hyperactivation of these signaling pathways in patients' cells was associated with elevated expression of ISGs and pro-inflammatory cytokines likely contributing to the diverse clinical manifestations observed. In two patients, Tofacitinib treatment successfully restored basal JAK/STAT activation levels of the JAK/STAT pathway resulting in clinical remission.

JAK1 conformation, particularly the closed one (inactive), is critical for its regulation (Glassman et al, 2022). By adapting AlphaFold2 we modeled human-JAK1 with a high confidence, providing the first closed conformation of human JAK1 to date. This model enabled to set a specific tool to screen potential GOF variant in *JAK1* by combining predicted pathogenic scores with structural consequences. As the identification of *JAK1* variants in patients with PIDD continues to rise (Horesh et al, 2024) this tool can help decipher the pathogenicity of these variants.

Further evaluation of the structural and energetic consequences of the five identified pathogenic variants suggested a loss of cis-regulation for the p.V464M, p.A634D, p.P815S, and p.R879G variants. The p.D1042Y and the p.N917D variants impacted both the activation loop and the ATP pocket-binding, which are also targeted by the kinase inhibitory region of SOCS1, a natural negative regulator of the pathway. Notably, the p.D1042Y is located at a known SOCS1 interaction site (Liau et al, 2018), indicating that these variants may result in hyperactivation through the loss of negative regulation. Thus, by leveraging AlphaFold2, our JAK1 model provided novel insights into GOF mechanisms exploited by *JAK1* pathogenic variants.

Consistent with previous reports of JAK1 GOF patients (Takeichi et al, 2022; Gruber et al, 2020; Horesh et al, 2024; Del Bel et al, 2017; Fayand et al, 2023) and murine models (Yasuda et al, 2016; Sabrautzki et al, 2013), the five patients in this cohort exhibited cutaneous manifestations of varying severity. Similar features have been observed in patients carrying *STAT6* GOF variants, potentially driven by an increased Th2 response through the IL-4/JAK1/STAT6 axis (Bao et al, 2013). Similarly, we found elevated IL-4 levels in the plasma of three patients, reminiscent with findings in patients with atopic dermatitis (Bao et al, 2013). In addition, we observed a decreased proportion of non-classical monocytes, Th1, and effector memory T cells. Given the observed inflammation in the skin, liver, and digestive tract in the analyzed patients, it is possible that these pro-inflammatory cells were infiltrating the affected tissues. Indeed, lymphocytic infiltrations of liver and dermis were previously observed in both patients and mice with hyperactivating *JAK1* variants (Takeichi et al, 2022; Sabrautzki et al, 2013; Yasuda et al, 2016; Del Bel et al, 2017).

In this study, all identified *JAK1* variants increased basal transcriptional activity of STAT1, STAT5, and STAT6 in the overexpression model. Therefore, it is not surprising that the biological and clinical features observed, such as autoimmunity, inflammation, elevated IgE, eosinophilia, and recurrent infections, overlap with those reported in patients carrying GOF variants in STAT1, STAT5, or STAT6 (Table 1) (Ott et al, 2023). Yet, most of the patients described here exhibited a broader spectrum of clinical manifestations than previously reported for JAK1 GOF variants (Takeichi et al, 2022; Gruber et al, 2020; Horesh et al, 2024; Del Bel

et al, 2017; Fayand et al, 2023). Notably, two patients presented with autoimmunity and one experienced recurrent viral and bacterial infections. As summarized by Ott et al, (Ott et al, 2023), the growing identification of patients with pathogenic variants in members of the JAK/STAT pathway has revealed a more comprehensive and expanded clinical profile. We observed converging clinical features between JAK1 GOF patients and those with various hyperactivating variants in the JAK/STAT pathway, likely reflecting the central, upstream role of JAK1 in cytokine signaling and activation of multiple STATs.

In the studied families, the variants exhibited incomplete clinical penetrance, a phenomenon increasingly recognized in autosomal dominant disorders (Gruber and Bogunovic, 2020). Similar incomplete penetrance has been reported for LOF variants in *SOCS1* (Hadjadj et al, 2020) and *PTPN2* (Jeanpierre et al, 2024), both negative regulators of the JAK/STAT pathway. This suggests that these variants act as predisposition factors to immune dysregulations with additional modifiers yet to be identified, likely influencing disease onset. Mechanisms such as gene–gene interactions, environmental triggers, and epigenetic regulation may contribute to this variability (Kingdom and Wright, 2022). Recent studies have revealed that many autosomal genes undergo random monoallelic expression (Stewart et al, 2025), including *JAK1* (Gruber et al, 2020), which could explain the incomplete penetrance observed. Of note, in our cohort, most of the carrier parents were healthy. This observation supported the hypothesis that disease manifestation could be linked to an altered expression ratio between the wild-type *JAK1* allele and the GOF variant toward the pathogenic variant. Unfortunately, we were unable to test this hypothesis further due to a lack of biological samples.

The functional characterization of these new GOF variants allowed for the distinction of two groups of GOF variants based on JAK1 and STATs activation: those with strong hyperactivation of the pathways (P815S, R879G, and D1042Y) and those with mild activation (V464M and N917D). These groups correlated with allele frequencies in the public database gnomAD. Indeed, the first group included private variants, similar to A634D (Del Bel et al, 2017) and S701I (Gruber et al, 2020), while the second group included variants already reported in gnomAD, similar to the four variants identified in the study by Horesh et al (Horesh et al, 2024). The identification of these new variants allowed to correlate the rarity of a given variant with its hyperactivation, rather than with the severity of the disease.

Understanding GOF mechanisms remained key to guide therapy as illustrated by the variable response to next-generation JAK inhibitors. These inhibitors, such as Filgotinib and Abrocitinib (Hu et al, 2021), were designed with higher specificity for JAK1 to reduce adverse events by selectively inhibiting hyperactivated JAK1. Yet, their limited efficacy in some cases could be attributed to cross-activation among JAK family members, requiring the inhibition of all affected JAKs to restore proper regulation of the JAK/STAT pathway (Gruber et al, 2020). In this cohort, two patients treated with Tofacitinib showed significant clinical improvement and resolution of symptoms. These findings underscore the importance of assessing the specific impact of each variant to tailor treatment strategies effectively. When considering JAK inhibitors, it could be crucial to assess the transactivation not only of JAK2 but also of JAK3 and TYK2. By targeting all relevant JAKs, a more

comprehensive therapeutic approach could be achieved, leading to better management of clinical manifestations associated with these variants. Finally, approved first-generation JAK inhibitors targeted the ATP-binding pocket. Variants in the kinase domain, particularly in the activation loop, could alter the binding of these inhibitors, rendering them ineffective. Therefore, the ectopic expression model remained a helpful tool to screen for the most effective JAK inhibitor.

In conclusion, we developed a structure-based predictive framework adapting AlphaFold2 to assess the pathogenicity of *JAK1* variants. Applying this approach to a cohort of patients with suspected PIDD led to the identification of novel hyperactivating *JAK1* variants and expanded the clinical spectrum of JAK1 GOF-associated disease to include autoimmunity and susceptibility to infections (Philips et al, 2022). Furthermore, our study highlights the importance to dissect GOF mechanisms to enhance our understanding of JAK1 kinase regulation and to identify the most effective JAK inhibitors.

# Methods

### Reagents and tools table

| Reagent/resource | Reference or source | Identifier or catalog number |
|---|---|---|
| **Experimental models** | | |
| U4C cells (*H. sapiens*) | Dr Sandra Pelligrini (Pasteur) | N/A |
| Lenti-X™293T cells | Dr Marianna Parlato | N/A |
| **Recombinant DNA** | | |
| pcDNA™6/JAK1-myc-HisA,B,C | Dr Delphine Cuchet-Lourenço and Dr Sergey Nejentsev (Cambridge University) | N/A |
| pLVX-EF1a-IRES-mCherry | Clontech, Mountain View, CA | N/A |
| pLVX-EF1a-JAK1-IRES-mCherry | This study | N/A |
| Luciferase reporter plasmid encoding a 4× STAT6 binding site | Sharma et al, 2024 | N/A |
| pGL4.52[luc2P/STAT5 RE/Hygro] | Promega | E4651 |
| pGL4[luc2P/GAS-RE/Hygro] | Promega | CS179301 |
| pGL4.45[luc2P/ISRE/Hygro] | Promega | E4141 |
| pTK-Green Renilla Luc Vector | Thermo Fischer Scientific | 16154 |
| packaging pMD2.G plasmid | Addgene | 12259 |
| VSV-G envelope expressing psPAX2 plasmid | Addgene | 12260 |
| **Antibodies** | | |
| CD3-PeCy7 | BioLegend | 300420 |
| CD4-BV510 | BioLegend | 317444 |
| CD8-BV711 | BioLegend | 344734 |
| CD19-APC | BioLegend | 363006 |
| CD14-PE | BioLegend | 301806 |
| pSTAT1-BV421 pY701 | BioLegend | 562985 |
| pSTAT5-BV421 pY694 | BioLegend | 562984 |
| Rabbit anti-pJAK1 pY1034/1035 | Cell Signaling | 74129 |
| Mouse anti-JAK1 | BioLegend | B610231 |
| Rabbit anti-pSTAT1 pY701 | Cell Signaling | 9167 |

| Reagent/resource | Reference or source | Identifier or catalog number |
|---|---|---|
| Rabbit anti-STAT1 | Cell Signaling | 14994 |
| Rabbit anti-pSTAT5 pY694 | Cell Signaling | 4322 |
| Rabbit STAT5 | Cell Signaling | |
| Rabbit anti-pSTAT6 pY641 | Ozyme | 56554 |
| Rabbit STAT6 | Cell Signaling | |
| Rabbit anti-GAPDH | Ozyme | 5174 |
| Goat anti-rabbit IgG | Cell Signaling | 7074 |
| Goat anti-mouse IgG | Cell Signaling | 7076 |
| Maxpar Direct Immune Profiling kit | Fluidigm | 201325 |
| **Oligonucleotides and other sequence-based reagents** | | |
| Primer forward mutagenesis c.2443 C > T: GCCAGTGACAtCATCATGTAAG | Eurofins | N/A |
| Primer reverse mutagenesis c.2443 C > T: CTGCACCGGCTTTCATAG | Eurofins | N/A |
| Primer forward mutagenesis c.3124 G > T: GGATGACCGGTACAGCCCTGT | Eurofins | N/A |
| Primer reverse mutagenesis c.3124 G > T: TTGACGGTGTAATACTCCTTATCG | Eurofins | N/A |
| Primer forward mutagenesis c.1390 G > A: GGGGATGTACATGCTGAGGTG | Eurofins | N/A |
| Primer reverse mutagenesis c.1390 G > A: TCCTCGCTTCCTTCTTGC | Eurofins | N/A |
| Primer forward mutagenesis c.2749 A > G: GAGTGGAGGTGACCACATAGC | Eurofins | N/A |
| Primer reverse mutagenesis c.2749 A > G: TCAGGCTTCAGAGATTTAAC | Eurofins | N/A |
| Primer forward mutagenesis c.2635 C > G: AAAGAGGATCGGTGACTTGGG | Eurofins | N/A |
| Primer reverse mutagenesis c.2635 C > G: AGGAAGCGCTTTTCAAAATG | Eurofins | N/A |
| Primer forward mutagenesis c.1901>T: TCCTCGCTTCCTTCTTGC | Eurofins | N/A |
| Primer reverse mutagenesis c.1901>T: ATTTCCCTGGACTTCTTCGAGG | Eurofins | N/A |
| Primer forward mutagenesis c.LOF: GGTGGCTGTTGCATCTCTGAAGC | Eurofins | N/A |
| Primer reverse mutagenesis c.LOF: TGCTCCCCTGTATTGTCC | Eurofins | N/A |
| ISG15 qPCR | ThermoFisher | Hs01921425_s1 |
| IFIT1 qPCR | ThermoFisher | Hs03027069_s1 |
| RSAD2 qPCR | ThermoFisher | Hs00369813_m1 |
| **Chemicals, enzymes, and other reagents** | | |
| Tofacitinib | Selleckchem | CP-690550 |
| Abrocitinib | Selleckchem | PF-04965842 |
| CEP-33779 | Selleckchem | |
| Legendplex Human Essential Immune Response Panel | Biolegend | 740930 |
| RNeasy Mini Kit | Qiagen | 74106 |
| QuantiTect Reverse Transcription Kit | Qiagen | 205313 |

| Reagent/resource | Reference or source | Identifier or catalog number |
|---|---|---|
| TaqMan PCR Master Mix | ThermoFisher | 4304437 |
| IL-2 | | |
| Hu-IFN-aA | Merck | IF007 |
| IFN-γ | | IMUKIN 2.10⁶IU |
| Lipofectamine™ 2000 | ThermoFisher | 11668019 |
| Dual-Glo assay | Promega | E2980 |
| Q5® Site-Directed Mutagenesis Kit | New England BioLabs | E0554S |
| PerFix EXPOSE | Beckman Coulter | B26976 |
| **Software** | | |
| OMIQ | https://app.omiq.ai/ | |
| Prism 10 | | |
| Biorender | | |
| NEBaseChanger | https://nebasechanger.neb.com | |

## Cohort of patients with suspected PIDD

The PIDD patients and their family members were recruited by specialists in clinical immunology across hospitals, mainly in France but also all over the world (including Argentina, UK, and Portugal). The broad inclusion criteria was the onset of an isolated or combined autoimmune disease before the age of 18 and included the following clinical diagnosis: ALPS-FAS (autoimmune lymphoproliferative syndrome): 12%, other ALPS: 5%, AI (autoimmune) cytopenia: 28%, AI hepatitis: 5%, Other organ-specific AI or poly-AI: 16%, pSLE (pediatric Systemic lupus erythematosus): 18%, JIA (juvenile idiopathic arthritis): 9%.

A minority of patients (including patient C1) were tested by targeted next-generation sequencing (NGS) with a PIDD panel, while all the other by whole-exome sequencing (WES).

Before the study, all patients signed informed consents approved by the CERAPH-Centre (IRB: #00011928). The biological samples are part of Inserm UMR1163/Imagine collection declared to the French Ministère de la Recherche (CODECOH no. DC-2020-3994). Consent was obtained from all subjects and the experiments were conformed to the principles set out in the WMA Declaration of Helsinki and the Department of Health and Human Services Belmont Report.

## Whole-exome sequencing

DNA was extracted from whole blood using standard methods. WES was performed on genomic DNA of patients using an AmpliSeq kit, with libraries analyzed on a Life Technologies Proton instrument. P3 was sequenced using SureSelect Human All Exon kit (Agilent Technologies) for targeted enrichment and an Illumina HiSeq 2000. Sanger sequencing was performed on DNA from patients and their parents to confirm the *JAK1* variants found by WES when samples were available. The reference sequence used for primer design and nucleotide numbering was *JAK1* (ENSG00000162434).

## Protein modeling and analysis

All methods are detailed in Appendix Supplementary Method 1.

## U4C cell culture

U4C cell line (JAK1⁻/⁻) was obtained from Dr Sandra Pellegrini and cultured in DMEM supplemented with 10% fetal bovine serum (FBS) and 1% penicillin/streptomycin (ThermoFisher Scientific). Cells were tested and free from mycoplasma contamination.

## Cloning and mutagenesis

Plasmid containing JAK1 cDNA: pcDNA™6/JAK1-myc-HisA,B,C plasmid was obtained from Dr Delphine Cuchet-Lourenço and Dr Sergey Nejentsev (Cambridge University). JAK1 cDNA was then subcloned into the lentiviral plasmid pLVX-EF1a-IRES-mCherry (Clontech, Mountain View, CA). Site-directed mutagenesis with specific primers, designed with NEBaseChanger (https://nebasechanger.neb.com) was performed using with Q5® Site-Directed Mutagenesis Kit (New England BioLabs #E0554S) according to the manufacturer's instructions. Supernatant containing lentivirus particles was generated from Lenti-X™293T cells co-transfected with transfer plasmid, packaging pMD2.G (12259; Addgene, Cambridge, MA) and VSV-G envelope expressing psPAX2 (12260; Addgene) plasmids using Lipofectamine 2000 (Invitrogen). The virus-containing medium was 0.45-mm-filtered and used to transduce cells in the presence of polybrene (4 mg/mL). mCherry-positive cells were sorted by fluorescence-activated cell sorting (Sony MA900 Cell Sorter) 5 days post-transduction.

## Luciferase reporter assays

U4C cells were seeded at $4 \times 10^4$ cells/well in a 96-well plate and, after overnight growth, transfected using 0.3 μL of Lipofectamine™ 2000 (ThermoFisher Scientific, #11668019) with JAK1 constructs (10 ng) together with firefly luciferase reporter plasmid (50 ng) under the control of the Interferon Stimulated Response Element (ISRE) promoter, Gamma Stimulated Signal (GAS) promoter, 5× STAT5 binding site or a 4× STAT6 binding site and a Renilla luciferase normalization vector (1 ng) for 6 h. Twenty-four hours post-transfection, cells were either left unstimulated or stimulated with IFN-α or IFN-γ $10^3$ U/ml during 6 h. Then, luciferase activity was measured using Dual-Glo assay (Promega, #E2980) and firefly luciferase activity was normalized against Renilla luciferase activity. Data are presented as normalized luciferase activity to unstimulated WT condition.

## STAT phosphorylation by flow cytometry on fresh whole blood

Fresh whole blood of patient or control were either left unstimulated or stimulated with IL-2 ($10^3$ U/ml), IFN-a ($10^3$ U/ml) or IFN-γ ($10^4$ U/ml) for 15 min at 37 °C. Extracellular staining was performed for 15 min during the stimulation process using anti-CD3-PeCy7; CD4-BV510; CD8-BV711; CD19-APC and CD14-PE (BD) antibodies. Cells were then fixed and permeabilized with the PerFix EXPOSE (Beckman Coulter, #B26976) kit according to the manufacturer's instructions. Intracellular staining was performed 2 h a room temperature using antibodies anti-pSTAT1-BV421 (pY701, BD #562985) or pSTAT5-BV421 (pY694, BD #562984). Cells were analyzed on the BD LSRFortessa™ X-20 SORP Cell Analyzer cytometer. CD4+ and CD8+ cells are gated on CD3+ cells. CD14+ cells are gated on CD3- and CD19- cells. The results were analyzed with FlowJo™ V.10.9.0 software.

## Immunoblot

Stably transduced U4C cells expressing JAK1 variants were either left unstimulated or stimulated with IFN-γ 1 U/ml for 15 min. For JAKinib treatments, cells were stimulated during 4 h with Tofacitinib (CP-690550, Selleckchem), Abrocitinib (PF-04965842, Selleckchem) or CEP-33779 (Selleckchem) at 0,1 or 1 µg/ml. Cells were washed and lysed in RIPA buffer with Halt protease and phosphatase inhibitors cocktail (ThermoFisher Scientific, #78444). Proteins were separated by electrophoresis on 8% Bis-Tris Bolt gel and transferred onto PVDF membrane (ThermoFisher Scientific, #IB24001). Membranes were blocked with TBS-Tween 0,1% and 5% BSA, incubated overnight with primary antibodies (1/1000$^e$) at 4 °C, then with secondary antibodies for 1 h at room temperature before revelation with pico plus reagent (ThermoFisher Scientific, #34580) and imaged using Chemidoc Imaging system (BioRad). The primary antibodies used were: pJAK1 (Cell signaling, pY1034/1035, #74129), JAK1 (BD, #B610231), pSTAT1 (Cell signaling, pY701, #9167), STAT1 (Cell signaling, #14994), GAPDH (Ozyme, #5174). The secondary antibodies used were goat anti-rabbit IgG (Cell Signaling, 7074) and goat anti-mouse IgG (Cell Signaling, 7076). Total proteins were blotted after stripping of the phosphorylated proteins.

## Interferon stimulated genes expression

RNA was extracted from PBMCs with RNeasy Mini Kit (Qiagen, #74106) and cDNA was generated using the QuantiTect Reverse Transcription Kit (Qiagen, #205313). Quantitative PCR was carried out with TaqMan PCR Master Mix (ThermoFisher Scientific, #4304437) on ViiA 7 Real-Time PCR System (Applied Biosystems). The expression of the following genes was quantified by RT-PCR: *ISG15, IFIT1, RSAD2, GAPDH* using TaqMan probes from Thermo-Fisher Scientific. Expression of each mRNA was normalized to the level of *GAPDH* by following calculation: $(2^{(Ct\ GAPDH-Ct\ gene)})*100$.

## Mass cytometry

### Samples staining

Immune phenotyping on thawed PBMCs ($3 \times 10^6$ cells) was carried out using the Maxpar Direct Immune Profiling kit (Fluidigm, #201325) with an antibody panel of 30 markers for CyTOF (Cytometry by Time Of Flight) analysis. Additional antibodies were added to detect FAS and certain immune checkpoint receptors (TIM3, TIGIT, ICOS, GITR, PD-1). The cells were incubated for 20 min at room temperature (RT) with 3 µL of heparin (Sigma-Aldrich, #H3149-10KU) at 10,000 U/mL and 5 µL of Human TruStain FcX (Biolegend, #422302), then incubated for 30 min at RT with the antibody cocktail for extracellular labeling. Blood lysis was performed using Cell Cal-lyse buffer (Thermofischer, GAS-010S100) according to the manufacturer instructions. Finally, cells were incubated in the Fix&Perm buffer (Fluidigm, #201325) with the Iridium intercalator at 1:1000 dilution (Fluidigm, #201325) overnight at 4 °C. Cell solutions were frozen at −80 °C prior to acquisition.

### Acquisition

Cells were washed and resuspended at a concentration of $1 \times 10^6$/mL in Maxpar Cell Acquisition Solution, a high ion concentration solution, and mixed with 10% EQ beads (allowing for calibration of

the automatic device) immediately before the acquisition. The acquisition of the events was carried out on the Helios mass cytometer 8 (Fluidigm) coupled with the CyTOF software version 6.7.1014 (Fluidigm) at the Pitie-792 Salpetriere Cytometry Platform (CyPS). The acquired data were normalized using the Fluidigm normalisation algorithm. Cells were selected by cell selection (Ir191+Ir193+), cell doublets were removed (Time/offset, Time/width, Time/Centre, and Time/residual) and dead cells were removed (Ir193+Rh103+). This selection is done automatically with the Pathsetter software.

### Data analysis

After control (number of cells per sample, expression pattern of all markers across samples), all cells were submitted to gating using The Maxpar Pathsetter analysis pipeline (Fluidigm). FCS files containing viable singlet cells were uploaded in OMIQ software, https://app.omiq.ai/.

## Cytokine assays

Plasma of controls or patients was collected after centrifugation of a heparinized blood sample. Cytokine concentrations were measured with the Legendplex Human Essential Immune Response Panel (Biolegend, #740930) according to the manufacturer's instructions. The decimal logarithms of the concentrations were normalized as follows: median concentration obtained in the 16 controls were defined as 0, the X-fold standard deviation above this median (0 to +4.5, coded in red) or below (0 to −2.5, coded in blue) was calculated for each individual cytokine.

---

### The paper explained

#### Problem
The growing number of *JAK1* variants identified in patients with auto-immune and autoinflammatory diseases presents a diagnostic and therapeutic challenge. Assessing the pathogenicity of these variants is essential to determine their clinical significance and guide targeted treatment strategies.

#### Results
A structure-based predictive framework adapting AlphaFold2 was developed to assess the pathogenicity of *JAK1* variants based on their impact on regulatory conformation. Dual-state modeling of 21,926 JAK1 variants enabled the distinction between pathogenic and non-pathogenic variants. Applying this approach to a large patient cohort led to the identification of five novel gain of function variants in key cis-regulatory and catalytic domains. Functional studies showed that these variants caused constitutive activation of JAK1 and downstream STAT1, STAT5, and STAT6 signaling, leading to elevated interferon-stimulated gene expression, increased pro-inflammatory cytokines, and immune cell imbalance. Treatment with pan-JAK inhibitors reversed these molecular abnormalities and improved clinical symptoms in two patients.

#### Impact
These findings broaden the clinical and mutational spectrum of *JAK1* gain of function syndrome and demonstrate the value of structure-informed variant analysis for identifying pathogenic variants. Functional characterization of individual variants can inform precision use of JAK inhibitors, supporting a tailored therapeutic approach for patients with immune dysregulation driven by JAK1 activation.

## Statistical analysis

Statistical analyses were performed using Prism 10 software. Specific tests applied to each experiment are detailed in the corresponding figure legends, with all significant *P* values displayed on the figures. No sample was excluded from the analysis. Studies performed in this manuscript were unblinded and no randomization procedures were applied.

## Data availability

Access to open and closed JAK1 models are available in ModelArchive (modelarchive.org) with the accession codes: https://www.modelarchive.org/doi/10.5452/ma-a1sai code 6umevq-XerO, and https://www.modelarchive.org/doi/10.5452/ma-l9b1f code zp13Kn2nWy, respectively. Clustering model script is available at https://www.ebi.ac.uk/biomodels/MODEL2507020001. WES data are available in the SRA database (accession number PRJNA1304256). Data for C1 patient will be available upon request, in accordance with the privacy and consent conditions of the collaborating center. CyTOF FCS files are available in the Zenodo database https://zenodo.org/records/16759495.

The source data of this paper are collected in the following database record: biostudies:S-SCDT-10_1038-S44321-025-00317-0.

## Peer review information

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

## Acknowledgements

This work was supported by the Institut National de la Sante et de la Recherche Medicale (INSERM) and by government grants managed by the Agence National de la Recherche as part of the "Investment for the Future" program (Institut Hospitalo-Universitaire Imagine, grant ANR-10-IAHU-01, Recherche Hospitalo-Universitaire, grant ANR-18-RHUS-0010), the Agence National de la Recherche (ANR-14-CE14-0026-01 "Lumugene"; ANR-18-CE17-0001 "Action"; ANR-21-CE17_0044 "Predict JIA"; ANR-22-CE15-0047-02 "BREAK-ITP") the Fondation ARC pour la recherche sur le CANCER, the Fondation pour la recherche Medicale (FRM: EQU202103012670). We thank NVIDIA® for computational support. We thank Dr Sandra Pellegrini, Dr Delphine Cuchet-Lourenço and Dr Sergey Nejentsev for provided materials.

## Author contributions

**Marie Jeanpierre**: Conceptualization; Data curation; Formal analysis; Investigation; Methodology; Writing—original draft; Writing—review and editing. **Orianne Debeaupuis**: Data curation; Software; Formal analysis; Methodology; Writing—original draft; Writing—review and editing. **Camille Brunaud**: Data curation; Methodology. **Judith Yancoski**: Data curation; Formal analysis. **Quentin Riller**: Formal analysis; Writing—original draft. **Jerome Hadjadj**: Investigation. **Marie-Claude Stolzenberg**: Methodology. **Giselle Villarreal**: Investigation. **Marie Martha Katsicas**: Investigation. **Mariana Villa**: Investigation. **Joao Farela Neves**: Investigation. **Jean-Louis Stephan**: Investigation. **Cedric Leonard**: Investigation. **Estibaliz Lazaro**: Investigation. **Jonathan Ciron**: Investigation. **Charlotte Boussard**: Investigation. **Fabienne Mazerolles**: Investigation. **Aude Magerus**: Investigation. **Olivier Pelle**: Methodology. **Cecile Masson**: Data curation; Formal analysis. **Yohann Schmitt**: Formal analysis. **Benedicte Hoareau**: Formal analysis. **Angelique Vinit**: Data curation; Formal analysis. **Bénédicte Neven**: Investigation. **Pierre Quartier**: Investigation. **Herve Isambert**: Methodology; Writing—original draft. **Matias Oleastro**: Investigation; Writing—original draft. **Silvia Danielian**: Investigation. **Marianna Parlato**: Conceptualization; Supervision; Investigation; Methodology; Writing—original draft; Writing—review and editing. **Frederic Rieux-Laucat**: Conceptualization; Supervision; Funding acquisition; Writing—original draft; Writing—review and editing.

Source data underlying figure panels in this paper may have individual authorship assigned. Where available, figure panel/source data authorship is listed in the following database record: biostudies:S-SCDT-10_1038-S44321-025-00317-0.

## Disclosure and competing interests statement

The authors declare no competing interests.

# Expanded View Figures

**Figure EV1. Validation of in silico pathogenicity prediction for *JAK1* variants.**

(**A**) Heatmap showing mean cluster values of in silico pathogenicity scores for *JAK1* missense variants. Red indicates higher pathogenic values or negative ΔΔG (protein destabilization), blue indicates lower pathogenic values or positive ΔΔG (protein stabilization). Actual mean values are displayed in each cell. Color scale represents normalized values for visualization (red to blue). (**B**) Histogram showing distribution of gnomAD variants ($n = 287$) across the two clusters. *P* value was computed with Bonferroni-corrected Chi-square test and shown in the figure. (**C**) ISGF3 complex (STAT1, STAT2 and IRF9) luciferase reporter activity in U4C cells co-transfected with a plasmid containing the Interferon-Stimulated Response Element (ISRE) and WT or JAK1 variants at baseline or after stimulation with IFN-α ($10^3$ U/ml for 6 h); $n = 3$. JAK1 variants are grouped on the basis of their cluster affiliation. Error bars represent standard error of the mean.

▶

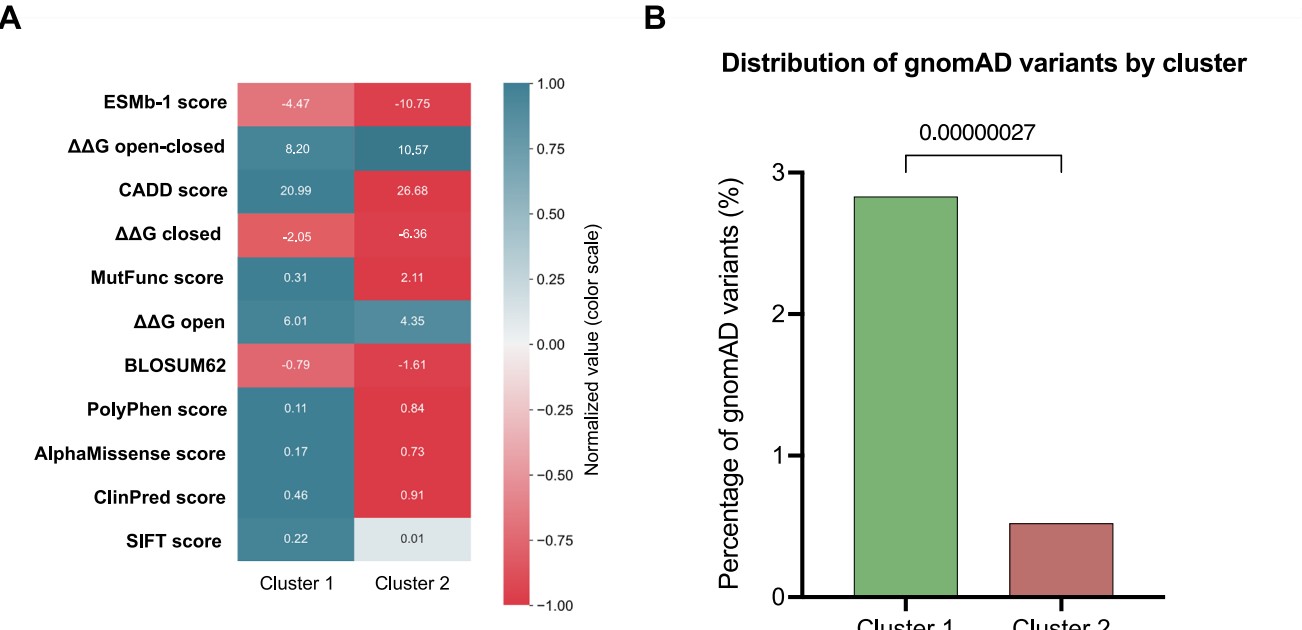

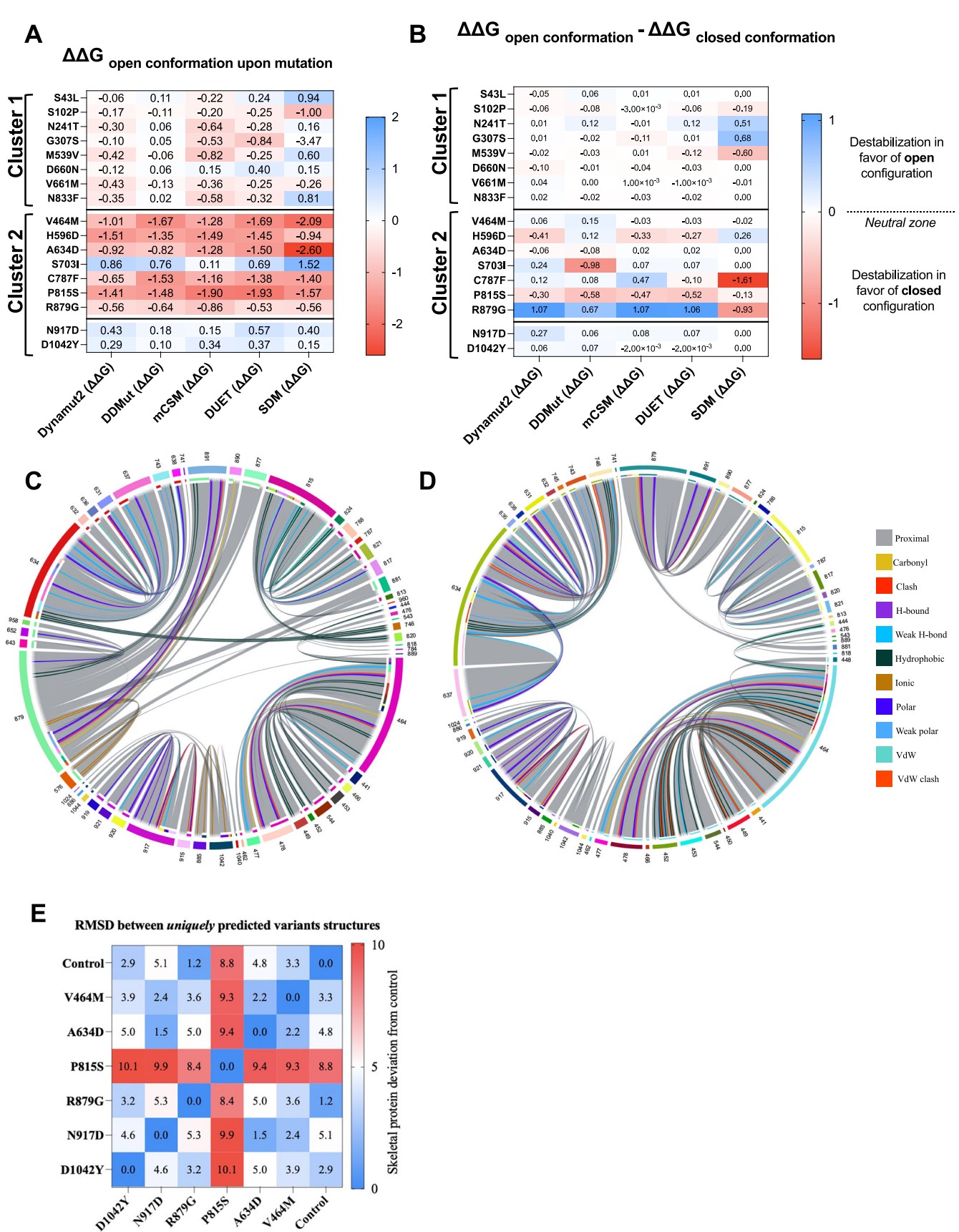

◀ **Figure EV2. Structural insights into uniquely predicted variants.**

(A) Predicted stability perturbation (ΔΔG) of JAK1 variants in the open conformation, computed using five different methods (see Appendix Supplementary Methods).
(B) Comparative analysis of ΔΔG values between open and closed JAK1 conformations, revealing the intrinsic tendency of mutated JAK1 to favor either conformation by minimizing the associated Gibbs free energy costs. Negative values indicate the tendency to adopt the closed conformation, while positive values point to a preference for the open conformation. (C, D) Chord plots illustrating residue-residue interactions in the JAK1 closed conformation. (C) Interactions involving wild-type residues.
(D) Interactions altered by the variant. Each thread represents an atomic interaction, color-coded by interaction type. (E) Heatmap of root-mean-square deviation (RMSD) computed between pair of structures, quantifying the similarity between two superimposed atomic coordinates (see Appendix Supplementary Methods).

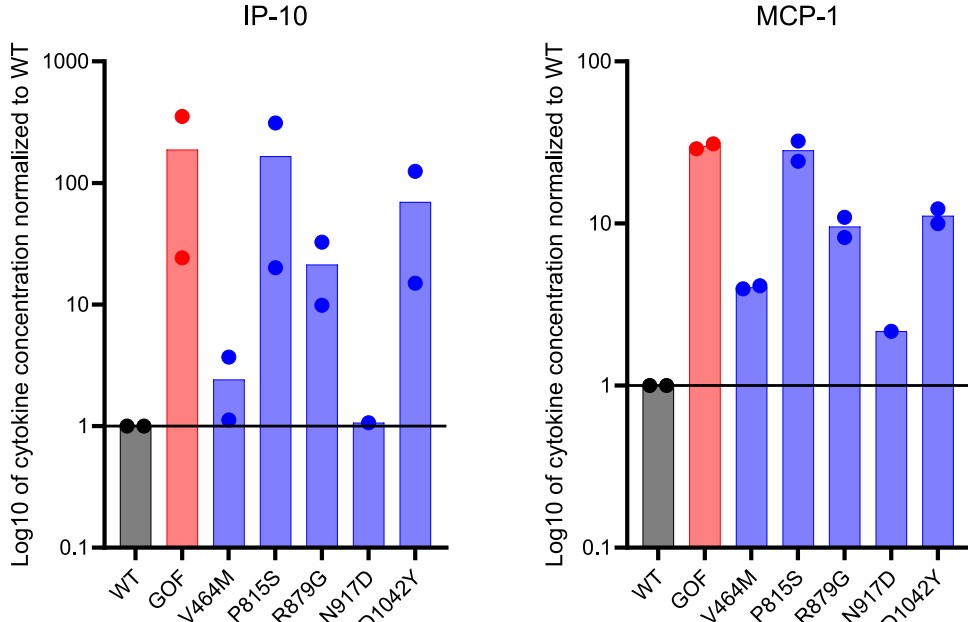

**Figure EV3. Inflammatory cytokines secreted by U4C cells expressing *JAK1* variants.**

MCP-1 or IP-10 levels measured in the supernatant of transfected U4C cells 24 h after transfection with WT, or JAK1 GOF (p.A634D), or patient's variants. $n = 2$ independent experiments.

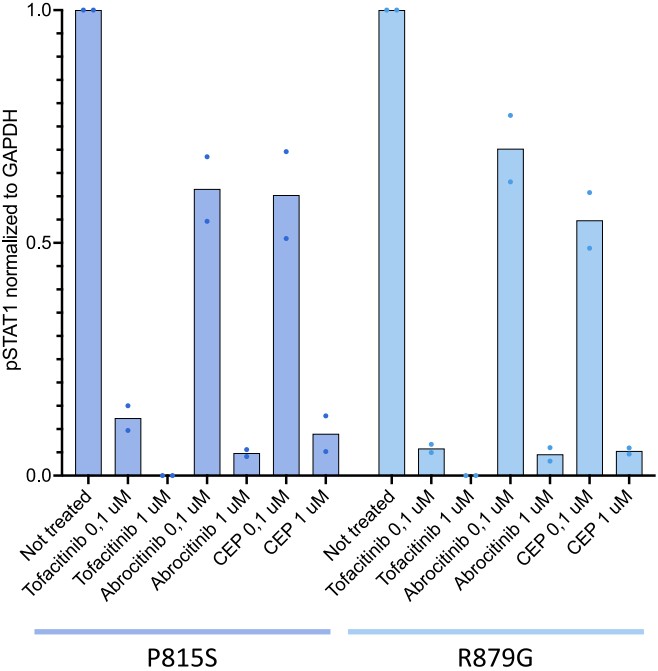

**Figure EV4. Differential effect of JAKinibs on transduced U4C cells.**

Western blot quantification of STAT1 phosphorylation in U4C cells transduced with different *JAK1* variants at baseline and after different JAK inhibitors treatment. pSTAT1 band quantification is normalized to total GAPDH and baseline condition. $n = 2$ independent experiments. Histogram represents the mean.

