## [Peer Review File · EMBO Molecular Medicine]

In silico modeling guides identification of novel JAK1 variants associated with immune dysregulation

Marie Jeanpierre, Oriane Debeaupuis, Camille Brunaud, Judith Yancoski, Quentin Riller, Jerome Hadjadj, Marie-Claude Stolzenberg, Giselle Villarreal, Marie Katsicas, Mariana Villa, Joao Neves, Jean-Louis Stephan, Cedric Leonard, Estibaliz Lazaro, Jonathan Ciron, Charlotte Boussard, Fabienne Mazerolles, Aude Magerus, PELLE Olivier, Cecile Masson, Yohann Schmitt, Benedicte Hoareau, Angélique Vinit, Bénédicte Neven, Pierre Quartier, Herve' Isambert, Matias Oleastro, Silvia Danielian, Marianna Parlato, and Frederic Rieux-Laucat

Corresponding authors: Marianna Parlato (marianna.parlato@inserm.fr) , Frederic Rieux-Laucat (frederic.rioux-laucat@inserm.fr)

Review Timeline:

Submission Date:	9th Apr 25
Editorial Decision:	28th May 25
Revision Received:	2nd Jul 25
Editorial Decision:	28th Jul 25
Revision Received:	3rd Sep 25
Accepted:	17th Sep 25

Editor: Lise Roth

Transaction Report:

28th May 2025

Dear Dr. Parlato,

Thank you for submitting your manuscript to EMBO Molecular Medicine, and please accept my apologies for the delay in getting back to you, as securing referees with the adequate expertise necessitated more time than usual, and one reviewer additionally requested a short extension to provide their report.

We have now received feedback from the two reviewers who agreed to evaluate your manuscript. As you will see from the reports below, they acknowledge the interest of the study and support publication of your work, pending appropriate revisions.

In order for us to consider the manuscript further, all reviewers' concerns must be addressed. EMBO Molecular Medicine only allows a single round of revisions, and acceptance or rejection of the manuscript will depend on how complete your responses are in the final version. If you would like to discuss further the points raised by the referees, I am available to do so via email or video. Let me know if you are interested in this option.

We are expecting your revised manuscript within three to four months, if you anticipate any delay, please contact us.

We require:

4) A .docx formatted letter INCLUDING the reviewers' reports and your detailed point-by-point responses to their comments. As part of the EMBO Press transparent editorial process, the point-by-point response is part of the Review Process File (RPF), which will be published alongside your paper.

5) A complete author checklist, which you can download from our author guidelines (<https://www.embopress.org/page/journal/17574684/authorguide#submissionofrevisions>). Please insert information in the checklist that is also reflected in the manuscript. The completed author checklist will also be part of the RPF.

6) All Materials and Methods need to be described in the main text using our 'Structured Methods' format. According to this format, the Methods section includes a Reagents and Tools Table (listing key reagents, experimental models, software and relevant equipment and including their sources and relevant identifiers) followed by a Methods and Protocols section describing the methods, ideally using a step-by-step protocol format. The aim is to facilitate adoption of the methodologies across labs. Please download and fill our Reagents and Tools Table template (.docx), which you can find in our author guidelines: <https://www.embopress.org/page/journal/14693178/authorguide#structuredmethods>.

7) Please note that all corresponding authors are required to supply an ORCID ID for their name upon submission of a revised manuscript.

8) It is mandatory to include a 'Data Availability' section after the Materials and Methods. Before submitting your revision, primary datasets produced in this study need to be deposited in an appropriate public database, and the accession numbers and database listed under 'Data Availability'. Please remember to provide a reviewer password if the datasets are not yet public (see <https://www.embopress.org/page/journal/17574684/authorguide#dataavailability>).

9) For data quantification: please specify the name of the statistical test used to generate error bars and P values, the number

(n) of independent experiments (specify technical or biological replicates) underlying each data point and the test used to calculate p-values in each figure legend. The figure legends should contain a basic description of n, P and the test applied. Graphs must include a description of the bars and the error bars (s.d., s.e.m.). Please provide exact p values.

10) Our journal encourages inclusion of *data citations in the reference list* to directly cite datasets that were re-used and obtained from public databases. Data citations in the article text are distinct from normal bibliographical citations and should directly link to the database records from which the data can be accessed. In the main text, data citations are formatted as follows: "Data ref: Smith et al, 2001" or "Data ref: NCBI Sequence Read Archive PRJNA342805, 2017". In the Reference list, data citations must be labeled with "[DATASET]". A data reference must provide the database name, accession number/identifiers and a resolvable link to the landing page from which the data can be accessed at the end of the reference. Further instructions are available at .

11) We replaced Supplementary Information with Expanded View (EV) Figures and Tables that are collapsible/expandable online. EV Figures should be cited as 'Figure EV1, Figure EV2' etc... in the text and their respective legends should be included in the main text after the legends of regular figures.

12) The paper explained: EMBO Molecular Medicine articles are accompanied by a summary of the articles to emphasize the major findings in the paper and their medical implications for the non-specialist reader. Please provide a draft summary of your article highlighting

13) Author contributions: CRedit has replaced the traditional author contributions section because it offers a systematic machine readable author contributions format that allows for more effective research assessment. Please remove the Authors Contributions from the manuscript and use the free text boxes beneath each contributing author's name in our system to add specific details on the author's contribution. More information is available in our guide to authors.

Please also suggest a visual abstract to illustrate your article as a PNG file 550 px wide x 300-600 px high. A cropped portion of this image will serve as thumbnail for the table of content on our webpage.

16) As part of the EMBO Publications transparent editorial process initiative (see our Editorial at <http://embomolmed.embopress.org/content/2/9/329>), EMBO Molecular Medicine will publish online a Review Process File (RPF) to accompany accepted manuscripts.

In the event of acceptance, this file will be published in conjunction with your paper and will include the anonymous referee reports, your point-by-point response and all pertinent correspondence relating to the manuscript. Let us know whether you agree with the publication of the RPF and as here, if you want to remove or not any figures from it prior to publication. Please note that the Authors checklist will be published at the end of the RPF.

I look forward to receiving your revised manuscript.

Yours sincerely,

Lise Roth

***** Reviewer's comments *****

Referee #1 (Comments on Novelty/Model System for Author):

I have some specific comments to the authors referring to the technical quality and the interpretation of their data.

Referee #1 (Remarks for Author):

The paper reports an improved, alphafold-based algorithm and a bioinformatic pipeline to model consequences of Jak1 missense mutations found in a cohort of patients with immune-related diseases. The computational analysis identifies five of the mutants as potential GOF variants. This assumption is tested by reconstituting JAK1-deficient cells and by analyzing cells from two patients carrying JAK1 GOF mutations. By and large the data confirm the GOF character of the mutant JAK1. Thus, the merit of this study is to identify new JAK1 GOF variants and to describe the clinical consequences. Furthermore, the improved algorithm and computational pipeline is shown capable of distinguishing JAK mutations with consequences for kinase activity from those that do not affect kinase activity. Finally, the study suggests that treatment with inhibitors is a viable strategy to reduce the effects of uncontrolled JAK activity.

A weakness of the paper is the use of the U4C cells for the analysis of JAK1 mutants as these cells were isolated after mutagenesis with a frameshift mutagen. In the times of CRISPR a better model could have been generated. Some of the inconsistencies described below may be the result of the limited usefulness of U4C cells.

Specific comments:

1. According to Fig. 3A the V464M and N917D mutations generate very little pSTAT1. The D1042Y mutant generates a bit more pSTAT1. In Fig. 4B V464M and N917D increase ISG levels, but D1042Y does not. Moreover, the reporter gene assays in 3B show transcriptional activity of D1042Y which is at least as high as that of V464M and N917D. According to these results the data linking JAK1 GOF with pSTAT1 and transcriptional activity of ISG are inconsistent.
2. The legend of Fig. 4B specifies that PBMC from 5 patients were analyzed. How is this reflected by the symbols in the bar graph? Are there single values for patients C1 and E1?
3. Patient C1 shows lymphoproliferation, but patient E1 does not. The values showing PBMC compositions of both patients in the bar graphs of fig. 4D are very similar. While I realize that the values are percentages, I find it surprising that the lymphoproliferation does not skew the relative cell counts. How is this interpreted?
4. The visual inspection of Fig. 5A suggests that pSTAT1 is incompletely suppressed with Tofacitinib. The blot should be quantified to yield an objective view. This request is relevant also in light of the fact that the authors ascribe a transcriptional impact to the slight elevation of pSTAT1 caused by some of the GOF mutants (Fig. 3). If so little pSTAT1 represents the transcriptional impact of the JAK1 GOF, than complete inhibition of this effect would be necessary for complete reversion by Jakinibs.

Minor:

- Introduction: refs 14 and 15 refer to the same publication
- Line 124: Garcia's study (ref 17) was published in 2022.
- Tofacitinib is designated as a pan-Jak inhibitor, but its ability to inhibit Tyk2 is very weak.

Referee #2 (Comments on Novelty/Model System for Author):

I like this paper and particularly the in silico predictions of GOF activity.

Referee #2 (Remarks for Author):

Review - Novel variants in JAK1 highlight different mechanisms of kinase dysregulation

This paper harnesses in silico modeling to assess the potential impact of novel JAK1 GOF variants. The in silico approach is a strong point as it identified five variants predicted to be pathogenic, developed an additional in silico tool for screening potentially damaging JAK1 variants, and developed the first closed-state model of JAK1. Some weaknesses became apparent in the latter half of the paper and are flagged herein.

Major Comments:

1. Clinical Narratives. The inheritance mechanism should be clearly stated. The cohort of patients displays heterogeneous clinical phenotypes, which is acceptable given the goal of phenotype expansion, however, this requires robust characterization of the phenotypes being presented. The authors' argument fails to strongly convince the reader of the causative nature of the variants, particularly due to the high rate of incomplete penetrance. Further connections should be drawn across patients, known JAK1 GOF cases, and other inborn errors of immunity of the JAK/STAT pathway. While these analyses are presented in the discussion, a brief comparison table could be added to strengthen this argument by highlighting shared clinical features. Furthermore, it is unclear how the cohort of 1590 patients was constructed. Please include details on inclusion and exclusion criteria, and distribution of diagnoses within your cohort. Consider a sensitivity analysis to support the proposed genotype-phenotype relationship. Finally, table EV1 appears to be shown twice in the submission.
2. Figure 3A Western Blot. This Western blot shows inconsistent loading, one GAPDH to represent multiple blots, and saturation of pSTAT1 long. The shape of the GAPDH blot suggests it is a different blot than the others. Please provide all representative blots for assessment. Furthermore, the text states "a trend towards increased JAK1 phosphorylation is observed at baseline for the p.V464M, p.N917D, and p.D1042Y variants, but only the only the p.P815S, p.R879G and GOF 260 variants show a significant increase (Fig 3A and Fig EV3A)". This is not clearly demonstrated in the blot, particularly for p.N917D, which appears lighter than the WT band. The quantification in EV3B does show a slight trend but this should be demonstrated on the representative blot. In addition, the quantification includes seven data points, but the legend for Figure 3A states it is representative of six blots.
3. JAK Inhibitors. This paper highlights the importance precise variant classification can guide effective JAKinib treatment. This is not clearly shown in the paper as both patients were treated with Tofacitinib which was not compared to other potential treatment approaches. They do discuss alternative, tailored approaches, although none are demonstrated experimentally, rendering it unclear why the importance of their model for guiding treatment is emphasized heavily.
4. In silico tool. I suggest that a major achievement of the paper is the in silico prediction work and specifically all the variants in Cluster 2. I suggest emphasizing this work as it will likely assist clinicians in the future who are confronted by variants of uncertain significance in JAK1.

Minor Comments:

5. Luciferase assay. This assay would benefit from the addition of one or more population variants as additional controls. This is not required for this submission; however, it should be considered for future studies.
6. Line 280. Change the phrasing of "probably predisposing to recurrent infections." Clearly state if the patient experienced recurrent infections and with what type of pathogen(s).
7. Line 400. "In this study, all identified variants led to basal STAT1, STAT5 as well as STAT6 transcriptional activity in overexpression model". The word "increased" was omitted (led to increased basal STAT1...).
8. Abstract. The cohort and patients should be mentioned within the abstract before description of the treatment with pan-JAK inhibitor. The specific pan-JAK inhibitor should also be mentioned in the abstract.
9. A new IUIS publication listing recognized IEs was just published in JHI. This is likely better than ref #1.
10. IEI is now typically used describe monogenic immune disorders, replacing PIDs or PIDDs. However, authors are free to choose their favorite acronym.
11. I wonder if the title might be modified to more accurately capture the breadth of work in this manuscript?
12. In Figure 1 the source of the MAF should be provided in the legend. I assume it is gnomAD v4?

***** Reviewer's comments *****

Referee #1 (Comments on Novelty/Model System for Author):

I have some specific comments to the authors referring to the technical quality and the interpretation of their data.

Referee #1 (Remarks for Author):

The paper reports an improved, alphafold-based algorithm and a bioinformatic pipeline to model consequences of Jak1 missense mutations found in a cohort of patients with immune-related diseases. The computational analysis identifies five of the mutants as potential GOF variants. This assumption is tested by reconstituting JAK1-deficient cells and by analyzing cells from two patients carrying JAK1 GOF mutations. By and large the data confirm the GOF character of the mutant JAK1.

Thus, the merit of this study is to identify new JAK1 GOF variants and to describe the clinical consequences. Furthermore, the improved algorithm and computational pipeline is shown capable of distinguishing JAK mutations with consequences for kinase activity from those that do not affect kinase activity. Finally, the study suggests that treatment with inhibitors is a viable strategy to reduce the effects of uncontrolled JAK activity.

A weakness of the paper is the use of the U4C cells for the analysis of JAK1 mutants as these cells were isolated after mutagenesis with a frameshift mutagen. In the times of CRISPR a better model could have been generated. Some of the inconsistencies described below may be the result of the limited usefulness of U4C cells.

We thank Referee #1 for their overall positive evaluation and for recognizing the novelty and strength of our improved computational pipeline in identifying functionally relevant JAK1 variants. Regarding the concern about the use of U4C cells: we acknowledge the limitations but would like to note that U4C cells are a well-established model in the field and have been previously and repeatedly used to investigate JAK1 function and signaling (Gruber et al. doi.org/10.1016/j.immuni.2020.07.006; Horesh et al. doi.org/10.1084/jem.20232387). Their complete deficiency in JAK1 provides a clean background for reconstitution studies, which was critical for our comparative functional assays. Nevertheless, we agree that future studies using CRISPR-generated models may provide additional insights.

Specific comments:

1. According to Fig. 3A the V464M and N917D mutations generate very little pSTAT1. The D1042Y mutant generates a bit more pSTAT1. In Fig. 4B V464M and N917D increase ISG levels, but D1042Y does not. Moreover, the reporter gene assays in 3B show transcriptional activity of D1042Y which is at least as high as that of V464M and N917D. According to these results the data linking JAK1 GOF with pSTAT1 and transcriptional activity of ISG are inconsistent.

We agree that the differences in pSTAT1 levels for these variants are not easily distinguishable in the blot provided in Figure 3A. To clarify this, we repeated the western blot with adjusted loading conditions and longer exposure times to enhance detection sensitivity, particularly at baseline. This allowed us to better visualize the bands for WT, V464M, N917D, and D1042Y. Quantification across eight independent western blots confirmed a consistent increase in baseline STAT1 phosphorylation for these three variants, supporting their classification as GOF mutations. This is further supported by the results in U4C

cells transduced with D1042Y, where we observed increased ISG expression consistent with a GOF phenotype here shown for the reviewer.

In contrast, the ISG signature shown in Figure 4B reflects measurements from patients' cells. Notably, patient E1 (carrying the D1042Y variant) was under corticosteroid treatment at the time of sampling. Thus, the apparent discrepancy can be explained by treatment-related effects rather than biological inconsistency.

2. The legend of Fig. 4B specifies that PBMC from 5 patients were analyzed. How is this reflected by the symbols in the bar graph? Are there single values for patients C1 and E1?

We thank the reviewer for this helpful remark aimed at improving clarity. For patients C1 and E1, the ISG signature was assessed in a single experiment, resulting in one data point each in the bar graph. In contrast, for the other patients, the experiment was performed multiple times, and the number of replicates is reflected in the number of symbols shown in the graph. We have revised the figure 4 legend to make this distinction clearer. (line 917-919)

3. Patient C1 shows lymphoproliferation, but patient E1 does not. The values showing PBMC compositions of both patients in the bar graphs of fig. 4D are very similar. While I realize that the values are percentages, I find it surprising that the lymphoproliferation does not skew the relative cell counts. How is this interpreted?

Patient C1 experienced episodes of lymphadenopathy at the age of 13, but the sample used for CyTOF analysis was collected later, at age 16, during a clinically stable period. Additionally, CyTOF was performed on thawed PBMCs, starting with 3 million cells per patient. Since the analysis does not derive from standardized volumes of whole blood, but rather from a fixed number of isolated PBMCs, the data reflect relative proportions within the PBMC compartment rather than absolute cell counts. As such, lymphoproliferation at the tissue level may not necessarily be reflected in the relative distribution of circulating immune cells in this type of analysis.

4. The visual inspection of Fig. 5A suggests that pSTAT1 is incompletely suppressed with Tofacitinib. The blot should be quantified to yield an objective view. This request is relevant also in light of the fact that the authors ascribe a transcriptional impact to the slight elevation of pSTAT1 caused by some of the GOF mutants (Fig. 3). If so little pSTAT1 represents the transcriptional impact of the JAK1 GOF, than complete inhibition of this effect would be necessary for complete reversion by Jakinibs.

To address this important observation, we tested a broader dose range of tofacitinib to assess the extent of STAT1 inhibition. Quantification from two independent western blots confirmed that higher concentrations of tofacitinib resulted in complete suppression of STAT1 phosphorylation. Furthermore, ex vivo assessment of STAT1 phosphorylation in PBMCs from patient C1 after tofacitinib treatment showed a strong reduction, restoring pSTAT1 levels to those observed in healthy controls. These results confirm the efficacy of JAK inhibition in reversing the transcriptional effects associated with JAK1 GOF mutations and are now shown in Figure 5A.

Minor:

- Introduction: refs 14 and 15 refer to the same publication

This has been now corrected in the revised manuscript.

- Line 124: Garcia's study (ref 17) was published in 2022.

Thank you for pointing this out. We have corrected the publication year of García's study to 2022 (line 133).

- Tofacitinib is designated as a pan-Jak inhibitor, but its ability to inhibit Tyk2 is very weak.

We have now revised the text to clarify that tofacitinib primarily inhibits JAK1 and JAK3, with weak activity against TYK2 (line 359-360).

Referee #2 (Comments on Novelty/Model System for Author):

I like this paper and particularly the in silico predictions of GOF activity.

Referee #2 (Remarks for Author):

Review - Novel variants in JAK1 highlight different mechanisms of kinase dysregulation

This paper harnesses in silico modeling to assess the potential impact of novel JAK1 GOF variants. The in silico approach is a strong point as it identified five variants predicted to be pathogenic, developed an additional in silico tool for screening potentially damaging JAK1 variants, and developed the first closed-state model of JAK1. Some weaknesses became apparent in the latter half of the paper and are flagged herein.

We thank Referee #2 for their positive evaluation of our work and for highlighting the strength of our in silico predictions and modeling approach. We appreciate the reviewer's constructive comments on the latter sections of the manuscript and have addressed the specific concerns in detail below.

Major Comments:

1. Clinical Narratives. The inheritance mechanism should be clearly stated.

We apologize for not clearly describing the mechanism of inheritance. In response to your comment, we have now included the genotypes in Figure 4A to provide clearer information. However, due to limited access to family samples, genetic segregation analysis was challenging for some cases.

The cohort of patients displays heterogeneous clinical phenotypes, which is acceptable given the goal of phenotype expansion, however, this requires robust characterization of the phenotypes being presented.

We appreciate this important point. To provide a more robust characterization of the diverse clinical phenotypes, we have expanded Table 1 to include additional detailed clinical information summarized from the reports in the appendix.

The authors' argument fails to strongly convince the reader of the causative nature of the variants, particularly due to the high rate of incomplete penetrance. Further connections should be drawn across patients, known JAK1 GOF cases, and other inborn errors of immunity of the JAK/STAT pathway. While these analyses are presented in the discussion, a brief comparison table could be added to strengthen this argument by highlighting shared clinical features.

As suggested, we have implemented table 1 now comparing the newly identified JAK1 GOF patients with previously reported cases, as well as with patients exhibiting hyperactivation of the JAK/STAT pathway due to other monogenic causes. Although the clinical spectrum is broad, this comparison highlights clear shared phenotypic features, thereby strengthening the argument for the causative nature of these variants.

Furthermore, it is unclear how the cohort of 1590 patients was constructed. Please include details on inclusion and exclusion criteria, and distribution of diagnoses within your cohort.

For over 25 years we have been studying patients with pediatric-onset autoimmune diseases. We initiated this cohort with the study of autoimmune lymphoproliferative syndromes (ALPS). In ALPS-FAS, in addition to non-infectious and non-malignant polyclonal lymphoproliferation, the main autoimmune manifestation is the occurrence of autoimmune cytopenias. In addition to the study of ALPS, we have therefore opened our studies to patients who initially present with autoimmune cytopenias (isolated or on at least two lines as in Evans syndrome). Some patients secondarily develop other autoimmune diseases, such as rheumatological diseases like lupus or juvenile idiopathic arthritis, or organ-specific diseases, whether autoimmune (T1D, AI hepatitis, enteropathy) or resulting from lymphocytic infiltration of tissues (granuloma-like or diffuse). Our study cohort therefore consists of more than 4,000 patients with these pediatric-onset autoimmune pathologies. The inclusion criteria are therefore: the onset of an isolated or combined autoimmune disease before the age of 18. Exclusion criteria are: adult onset, refusal of consent. Among this cohort of patients, we performed exome sequencing on 1,590 patients with the following distribution of pathologies:

ALPS-FAS: 12%

other ALPS: 5%

AI cytopenia: 28%

AI hepatitis: 5%

Other organ-specific AI or poly-AI: 16%

pSLE: 18%

JIA: 9%

The description of the cohort is now in Materials & Methods in the section PID cohort (line 477-490).

Consider a sensitivity analysis to support the proposed genotype-phenotype relationship. Finally, table EV1 appears to be shown twice in the submission.

Thank you for the suggestion to perform a sensitivity analysis to support the genotype-phenotype relationship. We fully agree that such an assessment is important for scientific validation. However, in the case of JAK1 GOF, only approximately ~25 out of over ~21,900 JAK1 variants currently have experimentally validated or phenotypically annotated data. Conducting a formal sensitivity analysis (via input perturbation, bootstrapping, or leave-one-out strategies) requires a sufficiently large and reliable set of ground-truth data to avoid spurious inferences. With only ~25 labeled variants available, we believe that performing such an analysis under these conditions would introduce significant bias and could lead to misleading conclusions about model performance or biological relevance. Therefore, to ensure accurate interpretation, we have refrained from conducting this analysis at this stage.

To support our position, we note that for a population of ~21,900 variants, estimating a sensitivity of 0.8 with a 95% confidence level and a ±5% margin of error would require at least 244 variants functionally assessed. This follows standard statistical principles for finite population sampling:

$$n = \frac{Z^2 \cdot p \cdot (1 - p)}{E^2} \cdot \frac{N}{N - 1 + \frac{Z^2 \cdot p \cdot (1 - p)}{E^2}}$$

Where:

n is the required sample size for conducting sensitivity analysis

N = 21,926 is the total number of substitution variants

E = 0.05 is the desired margin of error

Z = 1.96, corresponding to a 5% confidence interval

p = 0.8 (tunable parameter) is the expected sensitivity

We therefore respectfully maintain that deferring a sensitivity analysis until a larger, more comprehensive set of experimentally characterized JAK1 variants is available represents the most scientifically sound approach. We consider this an important future direction for this work.

2. Figure 3A Western Blot. This Western blot shows inconsistent loading, one GAPDH to represent multiple blots, and saturation of pSTAT1 long. The shape of the GAPDH blot suggests it is a different blot than the others. Please provide all representative blots for assessment. Furthermore, the text states "a trend towards increased JAK1 phosphorylation is observed at baseline for the p.V464M, p.N917D, and p.D1042Y variants, but only the only the p.P815S, p.R879G and GOF 260 variants show a significant increase (Fig 3A and Fig EV3A)". This is not clearly demonstrated in the blot, particularly for p.N917D, which appears lighter than the WT band. The quantification in EV3B does show a slight trend but this should be demonstrated on the representative blot. In addition, the quantification includes seven data points, but the legend for Figure 3A states it is representative of six blots.

Thank you for this important remark. The results previously shown in Figure 3A were obtained from two membranes loaded with identical samples, allowing us to present the pooled results. We acknowledge this was not clearly communicated; thus, we provide here for the reviewer the uncropped membranes that were originally part of Figure 3A.

However, we have now included a new representative blot which highlights better the mild GOF variants (V464M, N917D, D1042Y). To better visualize them, we adjusted the sample loading and overexposed the membrane to enhance detection of pSTAT1 signals at baseline. Quantification from eight independent experiments confirmed a consistent increase in STAT1 phosphorylation for these variants. However, JAK1 phosphorylation for these same variants was more variable and not consistently elevated. Given this, we have removed the quantification of JAK1 phosphorylation and attenuate statement referring to increased JAK1 phosphorylation at baseline for these variants to avoid overinterpretation. We also corrected the figure legend to accurately reflect the number of replicates included in the quantification. All original western blots used for quantification are now provided in the Source data.

3. JAK Inhibitors. This paper highlights the importance precise variant classification can guide effective JAKinib treatment. This is not clearly shown in the paper as both patients were treated with Tofacitinib which was not compared to other potential treatment approaches. They do discuss alternative, tailored approaches, although none are demonstrated experimentally, rendering it unclear why the importance of their model for guiding treatment is emphasized heavily.

To directly address this remark, we have now performed additional experiments comparing the effects of a JAK1-specific inhibitor (Abrocitinib), a JAK2-specific inhibitor (CEP), and Tofacitinib. Our data now show that Tofacitinib elicited the strongest inhibitory response, consistent with the clinical benefit observed in treated patients. We have added these data to better support the relevance of our model in informing therapeutic strategies in Figure 5A.

4. In silico tool. I suggest that a major achievement of the paper is the in silico prediction work and specifically all the variants in Cluster 2. I suggest emphasizing this work as it will likely assist clinicians in the future who are confronted by variants of uncertain significance in JAK1.

We thank and agree with the reviewer that the in silico predictions, especially the identification of Cluster 2 variants, are a major strength of our study. To support clinical utility, we have added a table listing all variants with their assigned cluster (Appendix Supplementary Table 1). While functional testing remains important, the strong concordance across the 12 tested variants highlights the model's potential to guide interpretation of uncertain JAK1 variants.

Minor Comments:

5. Luciferase assay. This assay would benefit from the addition of one or more population variants as additional controls. This is not required for this submission; however, it should be considered for future studies.

While not included in this study, we agree that including population variants in the luciferase assay would strengthen future analyses and will consider it in follow-up work.

6. Line 280. Change the phrasing of "probably predisposing to recurrent infections." Clearly state if the patient experienced recurrent infections and with what type of pathogen(s).

The phrasing on line 280 (now line 296) has been revised to clearly state that the patient experienced cellulitis of the thigh but the pathogen was not identified.

7. Line 400. "In this study, all identified variants led to basal STAT1, STAT5 as well as STAT6 transcriptional activity in overexpression model". The word "increased" was omitted (led to increased basal STAT1...).

Thank you for catching this omission. We have corrected the sentence to read "led to increased basal STAT1, STAT5, as well as STAT6 transcriptional activity." (line 415)

8. Abstract. The cohort and patients should be mentioned within the abstract before description of the treatment with pan-JAK inhibitor. The specific pan-JAK inhibitor should also be mentioned in the abstract.

The abstract has been updated to mention the patient cohort earlier (line 62-63) and now specifies that the treatment involved the pan-JAK inhibitor tofacitinib (line 67).

9. A new IUIS publication listing recognized IEIs was just published in JHI. This is likely better than ref #1.

We have updated reference #1 to cite the latest IUIS classification published in JHI (line 660).

10. IEI is now typically used describe monogenic immune disorders, replacing PIDs or PIDDs. However, authors are free to choose their favorite acronym.

We sincerely thank the reviewer for their comment and for leaving the choice of acronym open. Although the acronym IEI is widely used, we still believe it poses two important problems: 1- "Inborn" does not include somatic mutations that are increasingly described in immunological pathologies, and that can occur after birth and cause immune dysregulation. 2- We feel very uncomfortable to associate the term "error" with patients. For these two main reasons, we favor the acronym Primary Immune Dysregulation and Deficiency (PIDD).

11. I wonder if the title might be modified to more accurately capture the breadth of work in this manuscript?

Thank you for the suggestion regarding the title. We have revised it to better reflect the scope of the study, including both computational and clinical aspects and is now "*In silico* modeling guides identification of novel JAK1 variants associated with immunedysregulation".

12. In Figure 1 the source of the MAF should be provided in the legend. I assume it is gnomAD v4?

We have updated the legend of Figure 1 to specify that the MAF data are sourced from gnomAD v4.

28th Jul 2025

Dear Dr. Parlato,

Thank you for submitting your revised study. Referee #2 reviewed your responses to both referees, and as you will see below, this referee is overall satisfied with the revisions. I will therefore be able to accept your manuscript once the following editorial concerns are addressed:

1/ Referee's concerns:

Please address the remaining minor concerns raised by referee #2.

2/ Manuscript text:

- Please remove the red font text and only keep in track changes mode any new modification in the text.
- Please note that there is a discrepancy between "Benedicte Neven" in the manuscript and "B n dicte Neven" in our system; please ensure that the author names in the manuscript have all the accents and the correct spelling required for the final version; the email to contributing author Angelique Vinit (angelique.vinit@sorbonne-universite.fr) bounced.
- Methods:
 - o Human samples: please provide the full statement confirming that informed consent was obtained from all subjects and that the experiments conformed to the principles set out in the WMA Declaration of Helsinki and the Department of Health and Human Services Belmont Report. If collected and within the bounds of privacy constraints, please report on age, sex and gender or ethnicity for all study participants.
 - o Cell culture: please indicate whether the cells were authenticated and tested for mycoplasma contamination.
 - o Statistics: please provide a statement on sample size, inclusion/exclusion criteria, blinding and randomization
- Data availability section: please note that all large datasets should be deposited in public repository (including NGS, WES and mass cytometry) and an URLs should be provided. According to the journal's data policy, if practically possible and compatible with the individual consent agreement, we have to make sure that the authors deposit the human clinical datasets to public databases at the time of publication. Please remove "All the data are available upon reasonable request from the authors" and "all other data are available upon reasonable request from the authors".
- Acknowledgements: please note that the information provided should match the information entered in the submission system (currently, INSERM, the Centre de Reference Deficits Immunitaires Hereditaires (CEREDIH), ANR-21-CE17_0044 'Predict JIA, and the Fondation ARC pour la recherche sur le CANCER are not listed as funders in our system).
- "Disclosure and conflicts of interest" should be renamed "Disclosure and conflict of Interests Statement". Please remove "All co-authors have seen and agree with the contents of the manuscript".
- References: Please correct them to alphabetical order, with 10 author names listed before et al.

3/ Figures:

- EV figures should be uploaded as individual, high resolution figure files.
- Figures should be referenced in the chronological order in the manuscript text. Currently, Fig. 2A is called out before Fig. 1F; please correct.
- Western blots: please indicate if the membranes were stripped in the figure legends or methods.
- Several figure panels present standard deviations for n=2. Please remove and only keep the individual data points and the mean.
- Appendix table 1 should be renamed "Dataset EV1" and add a legend to the file.
- The movie files need their legends removed from the manuscript text and added in a simple text file zipped to each corresponding movie. The correct nomenclature is "Movie EV1" - EV3.
- Appendix: please add page numbers to the table of contents. You may wish to reformat and use a uniform font for the suppl. methods and case reports.
- Please address the queries from our data editors in the figure legends:
 1. Please note that the figure EV3, EV4 is mislabeled as figure EV4, EV5 respectively in the manuscript. This needs to be rectified.
 2. Please note that the exact p values are not provided in the legends of figures 3C, D; EV1 B
 3. Please note that information related to n is missing in the legends of figures 4E, 5C, D; S4
 4. Please note that the error bars are not defined in the legends of figures 4E, 5C, D; EV1 B, C; EV 3; S4
 5. Please note that the measure of center for the error bars needs to be defined in the legends of figures 3C-F; 4B, EV4

4/ Source Data:

- Please provide Source Data for Fig. 2A, F, G.
- Please provide the Source Data (WB) for Fig. 5A and the numerical raw data for Fig. 5C, D.

5/ Checklist:

- Cell materials: please make sure the subsection "Primary cultures" applies to your study, and fill in the subsection on

authentication and mycoplasma contamination.

- Human research participants: please fill in.
- Experimental study design and statistics: please fill in all subsections.
- Data availability: human clinical and genomic datasets should be deposited in a public access-controlled repository in accordance to ethical obligations to the patients.

6/ Synopsis:

- Please remove the synopsis from the manuscript text and upload it as a separate document.
- Please resize the visual abstract to 550px wide x 300-600px high and make sure the text remains legible. A cropped portion of this image will serve as thumbnail for the table of content on our webpage.

7/ As part of the EMBO Publications transparent editorial process initiative (see our Editorial at <http://embomolmed.embopress.org/content/2/9/329>), EMBO Molecular Medicine will publish online a Review Process File (RPF) to accompany accepted manuscripts.

This file will be published in conjunction with your paper and will include the anonymous referee reports, your point-by-point response and all pertinent correspondence relating to the manuscript. Let us know whether you agree with the publication of the RPF and as here, if you want to remove or not any figures from it prior to publication.

I look forward to receiving your revised manuscript.

Yours sincerely,

Lise Roth

***** Reviewer's comments *****

Referee #2 (Remarks for Author):

The authors have adequately addressed the major concerns presented by both reviewers for the manuscript titled "In silico modeling guides identification of novel JAK1 variants associated with immunedysregulation" (#EMM-2025-21786). Some suggested corrections are listed below.

- Title: There should be a space between immune and dysregulation
- Clinical narratives: It should be clearly stated that this is an autosomal dominant condition. Furthermore, the expansion of table 1 helps to understand the patient phenotypes, however, it would be beneficial to provide a brief summary of the most common clinical features, even if phenotypes are overall heterogenous. I would also like to see a brief summary of the similarities between these JAK1 GOF patients to those with other inborn errors of immunity of the JAK/STAT pathway rather than simply stating "converging clinical features between JAK1 GOF patients and those with various hyperactivating variants in the JAK/STAT pathway" (line 432-434).
- Line 275: "p.P815S, 275 p.R879G, and A634D GOF variants showed a clear pronounced increase (Fig 3A)". Remove "clear"
- Line 279-282: "exhibited a consistent, although modest, increase in STAT1 phosphorylation at baseline, reaching near statistical significance for p.N917D (Fig 3B), supporting their classification as GOF variants with lower activity". Remove "near statistical significance". This could be replaced with providing the p value, or removing altogether.
- Figure 5. The title "Tofacitinib treatment normalizes JAK/STAT pathway activation in patient-derived cells" implies each result in this figure is obtained from patient-derived cells. It should be rephrased to clarify that part of this figure (5A) was conducted in transduced U4C cells. The blots in 5A were representative of two experiments. In future, a triplicate would be more convincing.
- Suspected PIDD cohort. Would be more accurate to say that individuals with suspected PIDD were recruited to the cohort.

***** Reviewer's comments *****

Referee #2 (Remarks for Author):

The authors have adequately addressed the major concerns presented by both reviewers for the manuscript titled "In silico modeling guides identification of novel JAK1 variants associated with immunedysregulation" (#EMM-2025-21786). Some suggested corrections are listed below.

We thank the reviewer again for their valuable feedback, which has improved the clarity and quality

- Title: There should be a space between immune and dysregulation

- **We have corrected the spacing in the title so that it now reads "immune dysregulation."**

- Clinical narratives: It should be clearly stated that this is an autosomal dominant condition. Furthermore, the expansion of table 1 helps to understand the patient phenotypes, however, it would be beneficial to provide a brief summary of the most common clinical

features, even if phenotypes are overall heterogenous. I would also like to see a brief summary of the similarities between these JAK1 GOF patients to those with other inborn errors of immunity of the JAK/STAT pathway rather than simply stating "converging clinical features between JAK1 GOF patients and those with various hyperactivating variants in the JAK/STAT pathway" (line 432-434).

- **We now explicitly state that the condition is autosomal dominant in INTRODUCTION (lines 133). In addition, we have added the most recurrent clinical features, as suggested. Regarding the additional request to summarize similarities with other inborn errors of immunity in the JAK/STAT pathway, we felt this would be somewhat redundant within the same paragraph, as we have already described overlapping clinical features in detail earlier in the Results and Discussion sections.**

- Line 275: "p.P815S, 275 p.R879G, and A634D GOF variants showed a clear pronounced increase (Fig 3A)". Remove "clear"

- **As suggested, we have removed the word "clear". The sentence now reads: "p.P815S, p.R879G, and A634D GOF variants showed a pronounced increase (Fig 3A)..." (lines 290)**

- Line 279-282: "exhibited a consistent, although modest, increase in STAT1 phosphorylation at baseline, reaching near statistical significance for p.N917D (Fig 3B), supporting their classification as GOF variants with lower activity". Remove "near statistical significance". This could be replaced with providing the p value, or removing altogether.

- **We have now removed "near statistical significance" and we provide the exact p-value for p.N917D in the figure (line 295)**

- Figure 5. The title "Tofacitinib treatment normalizes JAK/STAT pathway activation in patient-derived cells" implies each result in this figure is obtained from patient-derived cells. It should be rephrased to clarify that part of this figure (5A) was conducted in transduced U4C cells. The blots in 5A were representative of two experiments. In future, a triplicate would be more convincing.

- **We have revised the Figure 5 title to clarify the experimental systems used. It now reads: "Tofacitinib treatment modulates JAK/STAT pathway activation in cells expressing JAK1 GOF variants." (line 1010)**

- Suspected PIDD cohort. Would be more accurate to say that individuals with suspected PIDD were recruited to the cohort.

- **As suggested, we have revised the text which now states: (lines 65, 177, 519, 526, 889) "**

17th Sep 2025

Dear Dr. Parlato,

Thank you for addressing the last editorial queries. I am pleased to inform you that your manuscript is accepted for publication and is now being sent to our publisher to be included in the next available issue of EMBO Molecular Medicine.

With kind regards,

Lise Roth
